

# The damability function: A probabilistic approach to regional landslide dam susceptibility analysis applied to the Oregon Coast Range, USA

Paul M. Morgan[1], alex grant[2], Will Struble[3], Sean LaHusen[2], Alison Duvall[1]

[1]Earth and Space Sciences, University of Washington, Seattle, 98195, USA
[2]United States Geological Survey, Earthquake Science Center, Seattle, 98195, USA
[3]Earth and Atmospheric Sciences, University of Houston, Houston, 77204, USA

*Correspondence to*: Paul M. Morgan (pmmorgan@UW.edu)

[**Abstract**]

Landslides can dam rivers and require rapid response to mitigate catastrophic outburst floods.
Here we present a workflow to map landslide dam formation susceptibility at a regional scale.
We define a probabilistic function that combines river valley width and landslide volume to
efficiently determine the likelihood of a landslide dam or 'damability'. We combine damability
values with landslide susceptibility to find landslide dam susceptibility. The valley width
measurements are automated using a new elevation threshold-based algorithm. Landslide volume
is represented as a statistical distribution from mapped landslides. We verify and apply our
approach to the Oregon Coast Range, USA and find high susceptibility in river headwaters and
generally steeper terrain, which in this case correlates with more resistant lithologies. We also
estimate volumes of the potential dammed lakes and find that most rivers with high dam
susceptibility are less likely to impound large lakes, because they have low drainage areas.
However, widespread susceptibility, and the critical potential impacts from exceptionally large
landslides, suggest this hazard should be considered in the Pacific Northwest. The damability
function workflow can readily ingest new data and can be applied more broadly to assess future
landslide dam hazards.

[**Short non-technical summary**]

When landslides dam rivers, the impacts can include catastrophic outburst flooding. This work
defines a function that combines river valley widths and landslide volumes to find the likelihood
that a river will be dammed by a potential landslide or 'damability'. We apply the method to the
Oregon Coast Range and find widespread high damability especially where rivers flow through
steep mountains with strong rocks. Our new workflow is flexible and can be applied more
broadly to other regions.






## 1 Introduction

Landslides are a widespread and destructive hazard present anywhere with steep slopes (Froude and Petley, 2018). The impact of these hazards can cascade from slope failure to
flooding when landslides intersect river valleys. Landslides can dam the flow of water and create a lake upstream of the slide deposit which may gradually flood roads or infrastructure. However, the greater danger lies in the possibility that the lake can break through the dam in a sudden outburst, draining rapidly and possibly catastrophically (Costa and Schuster, 1988; Fan et al., 2019; Korup and Tweed, 2007). Many historic landslide dam outburst floods have led to
widespread destruction and casualties (Costa and Schuster, 1991; Dai et al., 2005, 2021; Sattar and Konagai, 2012; Xu et al., 2009; Zeng et al., 2022). Landslide dams are potential hazards anywhere steep slopes abut rivers and especially in mountainous regions prone to landsliding (Costa and Schuster, 1988, 1991; Fan et al., 2020).

After a landslide dams a river, human intervention can prevent the rising lake waters
from breaching the dam in an outburst flood. The most straightforward method to prevent a large flood is the excavation of a spillway into the landslide dam (Sattar and Konagai, 2012), a technique that has been in practice for at least 500 years (Bonnard, 2011). Spillway excavation can both decrease the volume of the impounded lake, and thus the potential flood volume when the dam is overtopped, and can stabilize that flow, decreasing the peak breach discharge (Yan et
al., 2022). Timely coordinated landslide dam response by government agencies has averted disaster around the world (Bonnard, 2011; Duncan et al., 1986; Fan et al., 2020; Yang et al., 2010). However, most landslide dams fail within a short time after they form, with 50% failing within 10 days or less (Peng and Zhang 2012; Costa and Schuster 1988; Ermini and Casagli 2003). The combination of the importance of a response, and the short window for it,
necessitates pre-planning as a mitigation technique. The first steps are knowing which river stretches are susceptible to landslide dams, and where dams would be the most dangerous.

Several geomorphic indices have been proposed to estimate the likelihood of landslide dam formation (e.g. Dal Sasso et al., 2014; Ermini and Casagli, 2003; Fan et al., 2020; Korup, 2004; Tacconi Stefanelli et al., 2016; Wu et al., 2024). Fan et al. (2020) tested several of these
indices with four recent landslides from around the world and found that most of them performed adequately to predict landslide dam formation. Of these, the Morphological Obstruction Index (MOI) (Tacconi Stefanelli et al., 2016) is the simplest and the easiest to implement at landscape scales for situations where the properties of a future landslide must be inferred.

Within the MOI framework, dam formation or non-formation can be inferred from only
two parameters: landslide volume and valley width. Conceptually, large landslides impacting narrow valleys form landslide dams and, conversely, small landslides impacting wide valleys do not form landslide dams. A function we refer to as the damability function can estimate which landslide volume/valley width combinations will result in dam formation or dam non-formation for a specific region. This function is a modification of the formation volume and non-formation
volume equations from Tacconi Stefanelli et al. (2020). Damability functions can satisfactorily separate dam forming and non-forming landslides from a large historical database (Tacconi Stefanelli et al., 2015) and have been used to asses damming predisposition in Italy and Central Asia (Tacconi Stefanelli et al., 2020, 2023). However, the methods for calibrating or fitting a damability function to a new regional dataset are undefined and more widespread applicability of
the damability approach remains to be demonstrated.



Here we present a landslide dam susceptibility analysis focused on the Oregon Coast Range (OCR), United States of America. Tens of thousands of landslides have been mapped in the OCR (Burns and Madin, 2009), including 238 landslide dams (Struble et al., 2020, 2021). Additionally the OCR is subject to atmospheric rivers and intense precipitation events as well as strong shaking from Cascadia subduction zone earthquakes, all of which compound landslide and landslide dam hazards (Dettinger et al., 2018; Grant et al., 2022). We estimated landslide dam susceptibility by implementing a workflow based on a damability function, which we fit to a local landslide dam inventory. We used a new algorithm to automate valley width measurements. To estimate landslide volumes we used a large database of mapped landslide deposits to define a single empirical log-normal distribution of landslide volumes which is used across the study area. We also explored the utility of location specific landslide volume estimates using multivariate regression, however, we were unable to make satisfactorily predictions. Using the damability function, we found the geometric predisposition to damming, that is, the 'damability', of a valley. Then, we combined damability values with estimates of landslide susceptibility to find 'landslide dam susceptibility' values. Landslide dam susceptibility maps are therefore useful to planners (e.g. governmental emergency managers or transportation agencies) to identify places both prone to failure and damming. Finally, we estimated possible lake volumes for all potential dams to provide planners and other end users more information about the scale of potential impacts of future landslide dams. Our results provide a map of the landslide dam formation susceptibility and severity within the OCR. We find that landslide dams are most likely to form in river headwaters, high relief terrain, and more resitant lithologies. Most river stretches with the highest potential for landslide dam formation, may only impound relatively small lakes. However, the mapped widespread susceptibility and potential for the largest landslides to form dangerously large lakes that may require prompt mitigation, show that landslid dam hazards should not be overlooked across a broader region. We also investigate and discuss the damability function method, why it works, its flexibility, and its effectiveness for future evaluations globally.



## 2. The study area: Oregon Coast Range (OCR), USA

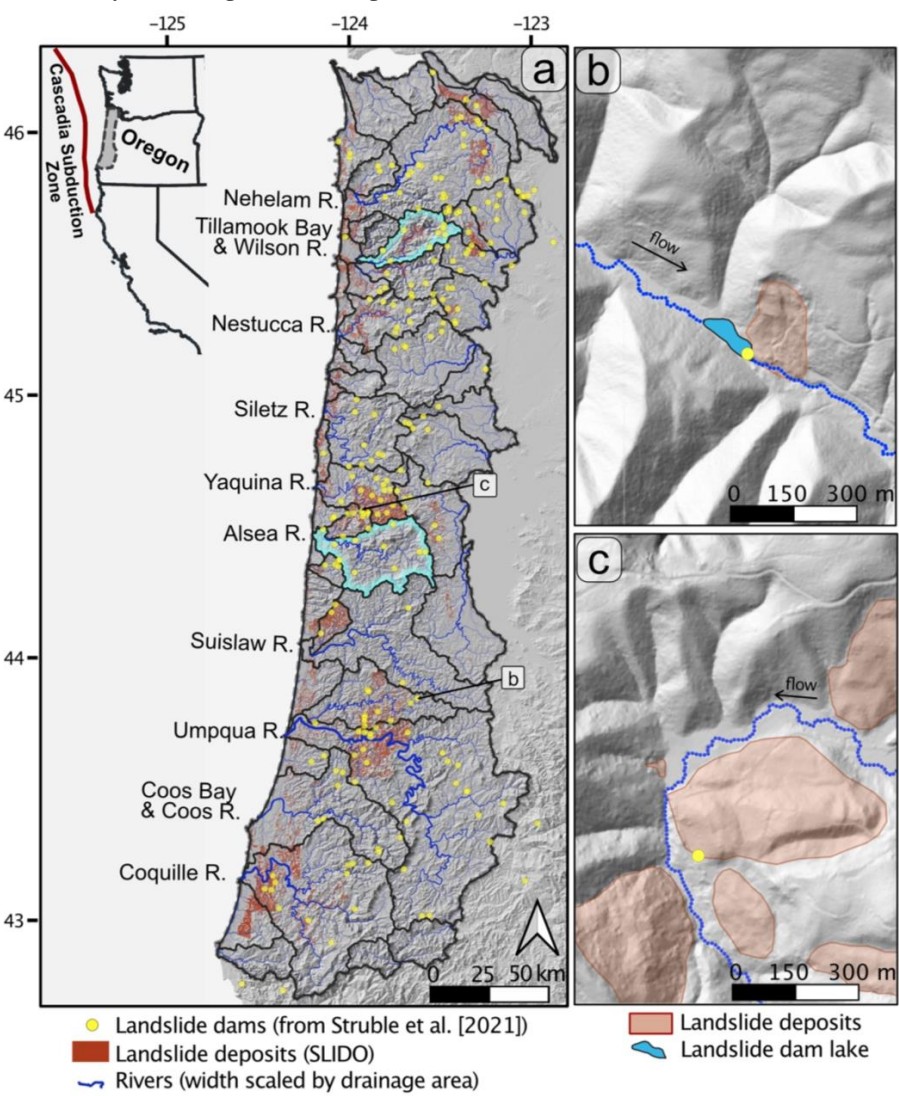


**Figure 1 a) overview of the Oregon Coast Range study area as a hillshade, derived river network, and landslide inventories. Oregon state-wide landslide Inventory (SLIDO, Franczyk et al. 2020) as red polygons, and Struble et al. [2021] landslide dam inventory as yellow dots. River drainage basins are delineated by black lines, with the Wilson and Alsea basins highlighted in light blue because they are discussed later (Figure 9, 10). b) and c) depict example landslides of different sizes**

**from the SLIDO inventory, and mapped landslide dam locations from Struble et al. [2021].**

The OCR is a roughly 60 km long mountain range along Oregon's Pacific coast, stretching from the mouth of the Columbia River in the north to the Klamath Ranges in the south (Fig. 1). These mountains are unglaciated and reach elevations of ~1200 m. The bedrock of the

range is mostly Eocene accreted volcanic terrains and marine sedimentary rocks, predominantly the siltstone and sandstone Tyee Formation in the central and southern areas (Lane, 1987; Roering et al., 2005; Wells et al., 2014). Conifer forests cover the landscape, which experiences





annual rainfall of 1.6 to 5.1 m (65-200 in.) per year, most of which during the winter and often in
the form of atmospheric rivers (Taylor, 1993).

130       Landslides are common in the OCR. During major winter storms, hundreds of shallow
landslides, which often mobilize into debris flows, have been observed (Robinson et al., 1999).
Widespread evidence of deeper-seated landslides exists across these hillslopes as well (LaHusen
et al., 2020). Shallow and deep landslides have been studied and inventoried in detail, including
in the Oregon statewide landslide inventory, or SLIDO (Franczyk et al. 2020 red polygons in

Fig. 1). Landslide dams have been documented in the area by Struble et al., (2021), who
published an inventory of 238 landslide dams (yellow dots in Fig. 1). These dams were primarily
caused by deep seated translational or rotational slides. Despite the proximity of the Cascadia
Subduction Zone, few dated landslide deposits or landslide dams can be correlated to
earthquakes (Grant, Struble, & LaHusen, 2022; Struble et al., 2021; LaHusen et al., 2020).

**3 Methods**

       Our approach to estimate landslide dam susceptibility along OCR rivers is outlined in
Fig. 2 and described in detail in the subsections below. First, we provide background on the
damability function approach (3.1). We measured valley widths across the study area using a
new algorithm based on elevation thresholds (3.2). We characterized the future landslide

volumes using a log-normal distribution of mapped landslide volumes (3.3.1). We also explored
using multiple regression to identify a spatially variable way to predict landslide volumes,
though ultimately this was not implemented (3.3.2). We created a new dam/non-dam landslide
dataset to calibrate a damability function (3.4.1). Then we fit this data to generate the OCR
specific damability function ($Damability_{OCR}$), and incorporated the regional log-normal landslide

volume distribution directly into the function to create the function ($Damability_{OCR-v}$) used in this
study (3.4.2). Lastly, we calculated damability, landslide dam susceptibility, and quantified the
magnitude of potential impacts by estimating the volume of landslide dam lakes through a proxy
(3.5).

       Our methods require the input of data from of a DEM, a general landslide inventory, a

landslide dam/non-dam inventory, and a landslide susceptibility map (Fig. 2). We used a Lidar
DEM from Oregon Department of Geology and Mineral Industries (DOGAMI) Lidar Program
Data. The SLIDO landslide inventory records mapped landslide deposit polygons within select
study areas throughout Oregon (Franczyk et al., 2020). We used all landslides in our study area
that were mapped following the 'special paper 42'(Burns and Madin, 2009) mapping protocols

(n>19,000) to characterize landslide volume. The landslide dam inventory of Struble et al. (2021)
records points where landslide deposits have dammed or currently dam rivers. We generated a
new landslide dam/non-dam inventory by incorporating remapped landslide deposits
corresponding to the Struble et al. (2021) points and by assessing a random selection of the
SLIDO landslide polygons. We then incorporated the Oregon State-wide landslide susceptibility

map, which was generated primarily using the SLIDO landslide inventory, DEM data, and
geologic data (Burns et al., 2016).





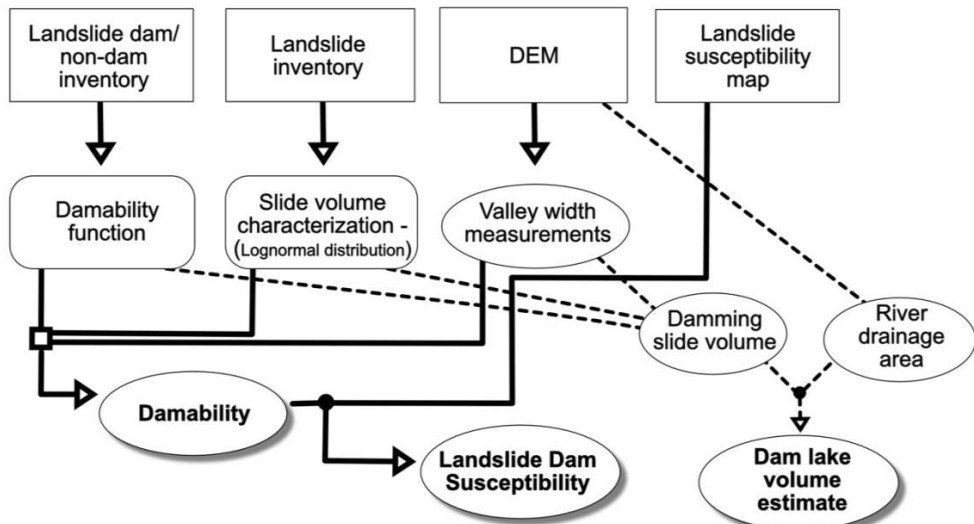

**Figure 2: Flow chart illustrating path from input datasets (square boxes) to interim results (rounded rectangles and ovals), to outputs (ovals with drop shadows). Rounded rectangles are values/equations that are constant across the study area, while ovals have unique values for each river point. Dashed lines represent the workflow for dam lake volume estimates, while solid lines represent the workflow towards landslide dam susceptibility.**

### 3.1 Damability function background

Landslide dam formation is a complex process that inherits all the geologic, and environmental uncertainties of landslide initiation and runout, as well as the discharge, erosive power, and flow path uncertainties of river valleys. It is difficult to know which hillslopes will fail in landslides, how large the landslides will be, and if the rivers will quickly transport the sediment or become (at least temporarily) impounded. Of metrics used to predict landslide dam formation, the 'annual constriction ratio', 'Dimensionless Morpho-Invasion Index', and 'Dimensionless Constriction Index' all require estimates of landslide velocity, which is difficult to measure for a known landslide, and even more difficult to estimate for an ancient or future landslide (Dal Sasso et al., 2014; Ermini and Casagli, 2003; Fan et al., 2020). Conversely, the global scale landslide dam formation susceptibility evaluation of Wu et al. (2024) bypasses landslide velocity or size. However, it relies on global river datasets that do not include rivers with sufficiently small drainage areas to capture and calibrate with landslide dams in the OCR.

Damability functions require only landslide volume and valley width, simplifying physically derived equations and complexity to two easily modeled terms that can be estimated and measured at landscape scales. Damability functions were introduced by Tacconi Stefanelli et al. (2016) using a large historical dataset including dam forming and dam non-forming landslides (Tacconi Stefanelli et al., 2015). They manually placed two functions in bi-logarithmic space that separate landslides into three domains, (the formation, non-formation, and uncertain domains) which they named Formation Volume and Non-Formation Volume equations (Fig. 3b). These damability functions define landslide volume (V) as a function of valley width ($W_V$) raised to a power, expressing how large a landslide needs to be to dam a river of a given valley width (Fig. 3b red and blue dashed lines respectively). This approach, can make analysis of landslide dam likelihoods difficult, requiring maps based on both formation and non-formation




volumes, and does not provide any additional uncertainty values other than a domain position. In this study we opted for a modified logistic damability function that collapses dam formation likelihood down to one value which includes the uncertainty in domain position. Specifically, we used a logistic regression to identify the damability function position and uncertainties based

on dam forming and non-forming landslides in our dataset.

We defined the damability of a river stretch as the likelihood that a river will be dammed if a landslide occurs there. This damability value is the dam formation likelihood output of the damability function. As described in the following sections, we assumed that the volume of the landslide is set by the region-wide landslide volume characterization and input this as a

lognormal distribution of landslide volumes into the damability function. A damability value predisposes (a) that a landslide occurs and intersects the valley with the measured valley width, (b) that the relationship between valley width and dam formation is defined by the damability function, and (c) that the volume of the landslide is statistically represented by the lognormal fit to the SLIDO inventory. Damability values range from 0 to 1 and are calculated at every point

where a valley width measurement is made.

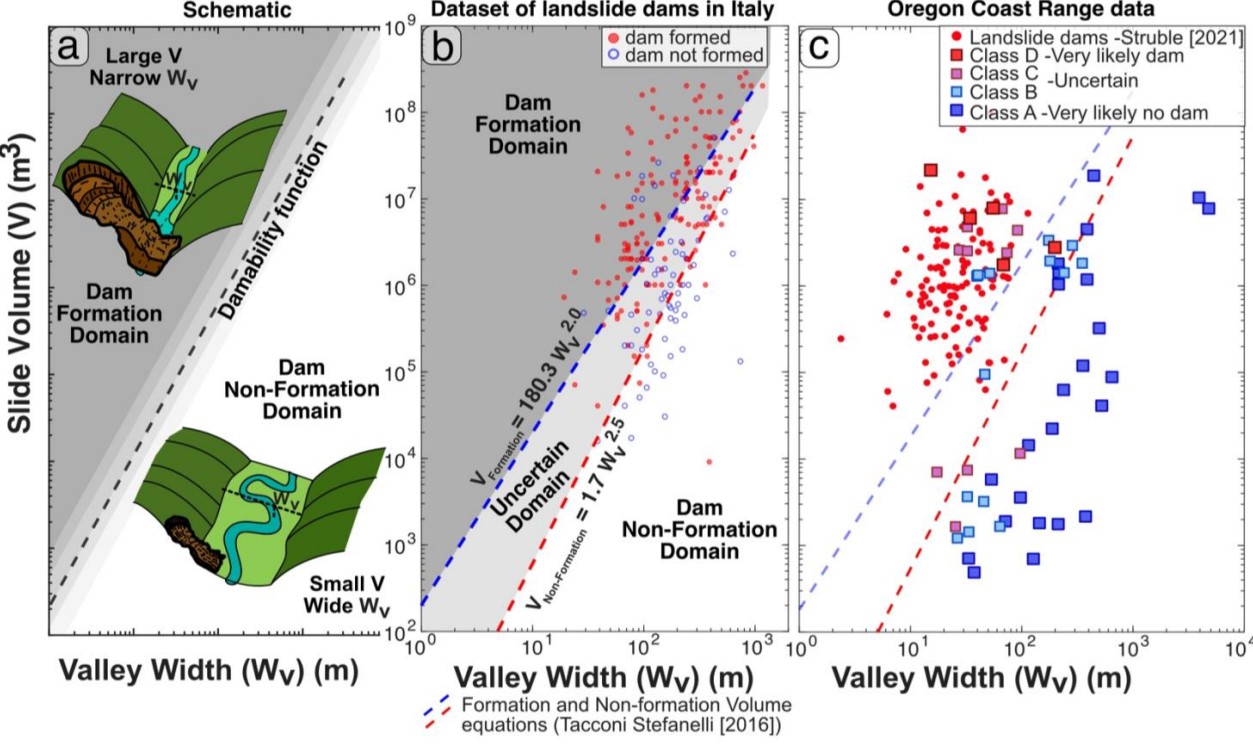

**Figure 3: Depictions of the damability functions for landslides in log-log valley width and landslide volume space. a) schematic diagram illustrating the form of the damability function. b) Reproduction of the dataset and functions presented by Tacconi Stefanelli et al. (2016). The formation volume domain function, in the blue dashed line is defined by the lack of dam-not-formed slides above it. The non-formation volume domain function, in the red dashed line, is defined by lack of dam-formed slides below it. c) Oregon Coast Range data (described in section 3.4.1.) Known landslide dams from Struble**



**et al. (2021) are plotted as small red dots, and the SLIDO (Franczyk et al., 2020) landslide subset in dark blue squares (very likely no dam), dark red squares (very likely dam) and light blue squares and light purple squares for slides with less certainty. Functions plotted in b) are plotted again in c).**

### 3.2 Automated valley width measurement

Several algorithms have been used to calculate river valley width, including landform mapping (Tacconi Stefanelli et al., 2020), topographic position index mapping (McMeckin, 2022), curvature wavelengths (Hilley et al., 2020), and threshold slope and elevation approaches (Clubb et al., 2022).  A threshold approach involving the elevation of valley walls above the river has the most physical connection to the pooling of water from river-flow impoundment. We

implemented a novel threshold-elevation based approach to calculate valley width using TopoToolbox functions (Schwanghart and Scherler, 2014), which we outline in Fig. 4.  The advantages of our approach when compared to other methods include: repeatability in the TopoToolbox coding infrastructure (rather than restriction to a GIS program), simplicity of the geometric calculations with no required complex landform classification, and the ease of use in

comparison to Python based codes available (e.g., Clubb et al., 2022).

   Our workflow starts with identifying rivers using a flow routing algorithm imposed on a 2m resolution lidar DEM of the study area, which is publicly available from the Oregon Department of Geological and Mineral Industries. While we used a detailed 2m resolution DEM in this study, the valley width algorithm performs well on DEMs with resolutions as coarse as

10m. We demarcated rivers based on a minimum drainage area cutoff of 2.25 km$^2$, which balances computational feasibility while capturing the prominent drainage area positions of the mapped landslide dams in the study area (Struble et al., 2021).

   Our algorithm categorizes a pixel as part of the valley if its elevation is less than a defined threshold elevation (10 m) above the nearest river pixel along a flow path. Profiles are

taken perpendicular to the river (Using TopoToolbox's swath profile extraction) to measure the distance that is within the valley. When the river does not flow parallel to the valley centerline, this results in profiles that record widths that are too large, but when the river flows parallel to the valley, the recorded width will be correct and the minimum possible recorded width. Our algorithm captures the minimum value along a moving window (100 m radius in this study) that

exceeds rivers meander wavelength, which is generally <150 m where river meanders don't create their own valley walls. While the elevation threshold choice alters the magnitude of the valley width measurements, it does not affect the relative spatial distributions (Supp. Fig. S1). A threshold elevation of 10 m was found to match valley widths previously hand measured in a catchment within the study area (May et al., 2013)(Supp. Fig. S2). Using this approach, valley

width measurements were extracted from the river network every 100 meters for input into the landslide dam susceptibility analysis.

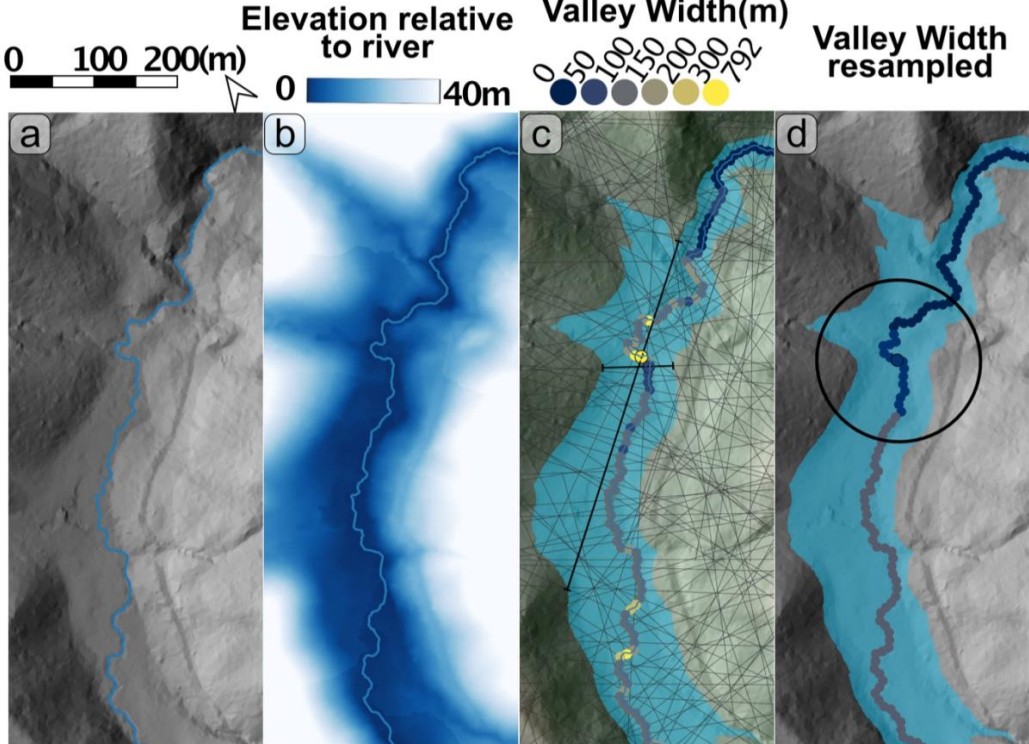

**Figure 4: Overview of valley width methods. a) hillshade of a sample valley, with the extracted river plotted, b) white to blue colour scale representing elevation above the river, along a flow path, c) valley extraction plotted as a blue polygon, light grey lines represent profile lines perpendicular to the river path, and river points are coloured by how much of their profile line lies in the valley. Two black profile lines and outlined river points, illustrate anomalously high values where the river path is perpendicular to the valley, and accurate low values where the river is parallel to the valley. d) final valley width result, which is the minimum value within a 100 m radius moving window, the size of which is plotted as a black circle.**

## 3.3 Landslide Volume Modeling

### 3.3.1 Empirical estimates of landslide volume

The size of future landslides can be estimated numerically based on physical laws and simulation of slope failure on the terrain (Bellugi et al., 2021), or empirically by using existing volume or area distributions from landslide inventories (Lombardo et al., 2021). We developed an empirical volume distribution for mapped landslides that we then used to estimate landslide dam formation likelihood. We opted for the empirical approach because 1) it is computationally expensive to simulate a landslide on every slope in our study area and 2) an empirical approach may be more likely to account for the wide range of possible landslide sizes observed in our study area. Whereas power law scaling (Tebbens, 2020) and similar alternative functions (Stark and Hovius, 2001) have been used to characterize landslide size distributions in the past, we choose a log-normal distribution. Medwedeff et al. (2020) demonstrated that log-normal functions can fit landslide size inventories, and that they capture absolute characteristic landslide sizes, while power laws only capture relative frequencies. Log-normal functions are also advantageous for working with statistical models (Bryce et al., 2022; Lombardo et al., 2021; Moreno et al., 2022).



We treated landslide volume estimation as a constant single statistical distribution across the study. We used the normal distribution of the log (base 10) of the landslide volume distribution captured from the SLIDO inventory (Fig. 5), defined by a mean ($\mu$) of 4.44 and
standard deviation ($\sigma$) of 1.25 (or ~28,000 m$^3$ plus or minus one standard deviation). The volumes are estimated by the mapper following the 'Special Paper 42' landslide mapping protocols, (96% of inventory has volumes) where the mapper estimates or measures the slide depth and multiplies that by the slide area (Burns and Madin, 2009). A comparison of the log-normal fit and data distribution is visible in Fig. 5.

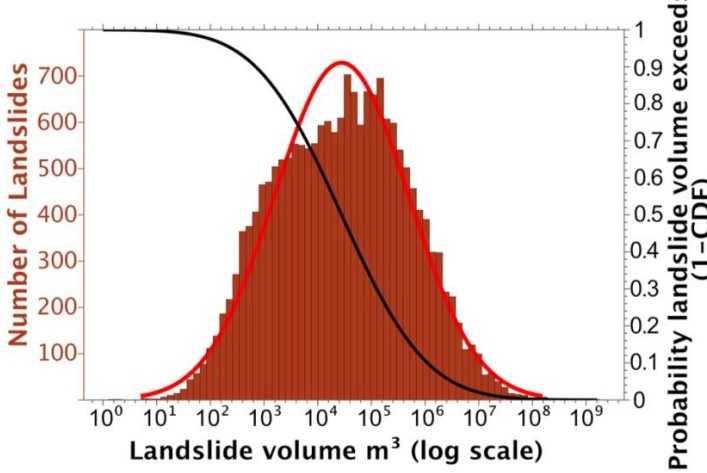


**Figure 5:** **Histogram of the base 10 logarithm of landslide volumes from the SLIDO landslide inventory (Franczyk et al., 2020). The log-normal fit is plotted in red, and the probability that a landslide is greater than a certain volume, or one minus the cumulative distribution function (cdf), in black.**

**3.3.2 Spatially variable estimation of landslide volume (not implemented in final workflow)**

The characteristic size of landslides may not be constant across the OCR. Variations in local lithology, and structure may impact the style or size of landsliding (LaHusen and Grant, 2024), and possibly also feedback into the valley widths. With these concerns in mind, we
attempted to make location/hillslope specific landslide volume predictions using statistical regressions. A few studies have demonstrated probabilistically that landslide deposit area may depend on local topographic morphology (Lombardo et al., 2021; Moreno et al., 2022; Qiu et al., 2018). Lombardo et al. (2021) and Moreno et al. (2022) implemented versions of a generalized additive model (GAM) to predict the maximum landslide surface area within a given slope unit
using a set of predictor variables including: relief, slope, roughness, and earthquake shaking. Both studies trained the models with data from earthquake triggered landslide inventories and determined that their data-driven statistical modelling approaches are relatively effective. To consider if such relationships hold for the OCR, we used several methods to create a statistical model using local landscape parameters to estimate local landslide volume. The parameters we
used included: relative relief, slope (mean and range), profile curvature (mean and range), vector ruggedness measure (mean and range)(Sappington et al., 2007), terrain ruggedness index (mean and range)(Riley et al., 1999), and geologic rock type. We extracted these variables using both a 500 m radius moving window and slope unit delineations following the input parameters of





Moreno et al. (2022) and the algorithm of Alvioli et al. (2016). We trained statistical models with volumes recorded from an 80% training dataset of the SLIDO inventory. The regression models include general linear models (GLM) (using linear and quadratic n=4 functions for each predictor variable), and general additive models (GAM).

### 3.4 Region specific damability function fitting

### 3.4.1 Function calibration data development


To identify a region specific damability function, that can be used within the OCR, we populated the bi-logarithmic slide volume/valley width parameter space (Fig. 3c) with local landslides that both formed and did not form landslide dams. We used an inventory of 239 formed landslide dams from Struble et al. (2021) as we are confident that landslides caused dams
at these locations. Identifying landslides that entered river valleys but did not form dams is less straightforward to assess without a long historical landslide dam inventory. To populate this part of the parameter space, we analyzed 100 landslides from a subset of the SLIDO inventory: 50 random landslides and the 50 landslides with the largest surface area. None of these landslides currently dam rivers, so we assessed them for evidence of past dam formation. Signs of past dam
formation include: an arcuate toe that extends well into river valley, or is cut off before reaching other side of the valley, an inner gorge where landslide intersects the river, and evidence of aggradation upstream of possible dam location. Assessing past landslide dam formation is subjective, thus we categorized each landslide deposit with a four part (A-D) Past Dam Formation Assessment (PDFA) class, where a PDFA-class A landslide very likely did not form a
dam, a PDFA-class D landslide very likely did form a dam, and PDFA-class B and PDFA-class C represent intermediate points where the inference of past dam formation is uncertain (Supp. Fig. S3). For each landslide used in the calibration, we remapped the surface area expression of the landslide, converted the area to volume using the area-volume scaling relationship of Larsen et al. (2010) for bedrock landslides, and manually mapped the valley width downstream of the
landslide.

### 3.4.2 OCR dam formation domain model fitting

We modeled the likelihood of dam formation from observed dam and non-dam forming
landslides as a modified logistic function (Eq. 1) using the ratio of estimated landslide volume (V) to measured valley width (Wv). Equation 1 is the generalized form of the damability function (*Damability$_{General}$*).

$$Damability_{General} = \frac{1}{1+e^{-k(\frac{log_{10}(V)}{log_{10}(aW_V)} - X_0)}} \qquad (1)$$


Where $k$, $X_0$, and $a$ are fitted parameters related to curve steepness, the expected midpoint value, and scaling factor between valley width and volume, respectively. Predicted values of damability range from zero to one, reflecting the likelihood that a given combination of landslide volume and valley width would form a landslide dam. The damability function was fit using nonlinear
least squares regression using the SciPy curve_fit regression tools for Python (Virtanen et al., 2020). With a damability function established we can find the likelihood that a valley width and landslide volume pair will dam a river.





### 3.5 Dam susceptibility, and dam lake volumes


We incorporated the likelihood of landslide occurrence into a metric we refer to as landslide dam susceptibility, by using the statewide map of landslide susceptibility (Burns et al., 2016). The statewide map breaks down susceptibility into 4 categories: 'low' (1), 'moderate' (2), 'high' (3), and 'very high' (4), where the category is determined by landslide inventory analysis, geology, and slope; except for the 'very high' category, which is determined by the location of mapped landslides. We normalized these values, with 'high' and 'very high' areas given a value


of 1, 'moderate' areas given a value of 0.5, and 'low' areas given a value of 0.25. This normalization was chosen conservatively to reflect that landslides still occur in the low and moderate zones. Of the test landslides in Burns et al. (2016), 65% were in 'high' areas, 16% in 'moderate' and 5% in 'low'. Landslide dam susceptibility is defined by the damability multiplied by the normalized landslide susceptibility values, and ranges from 0 to 1.


To estimate the possible lake volume arising from a future landslide dam at each river point, we used a scaling relationship developed by Argentin et al. (2021) through simulated landslides on alpine rivers. This proxy relationship calculated from only landslide volume and river drainage area was found to account for 60% of the variation in modelled lake volumes (EQ. 7 in Argentin et al., 2021). At each river point we input the measured drainage area and a


landslide volume corresponding to the minimum volume needed to likely dam a river of that width. The minimum dam forming landslide volume is calculated as the landslide volume where the damability value is 0.5 ($V_{50}$ -Eq. 3). Although these relationships were developed using alpine river geometries, they are likely still a good precursory estimate of the possible volumes of landslide dam lakes.


### 4 Results

### 4.1 Landslide volume characterization

Figure 5 presents the lognormal fit to the volumes from the >18,000 landslides in the SLIDO inventory. This dataset is well represented by a lognormal statistical distribution. We use


this single lognormal distribution as an input for landslide volume estimates across the study area where we calculate damability in this study. This statistical distribution is inserted directly into the damability function described in section 4.2 (Eq. 4).

We use the same volume distribution across the study area because the location specific estimates are unsuccessful. Figure 6 shows our best fitting model predictions for landslide


volume based on location properties (e.g. relief, slope, roughness, and lithology) compared to measured landslide volumes. In a successful statistical prediction, we would expect the points to extend along the 1:1 line without systematic errors. Instead, we find that predicted volume values are significantly different than observed volumes (Fig. 6). Model result errors are presented in Table 1. Based on these results for the SLIDO inventory, we are unable to justify using these


statistical regression techniques to predict local landslide volume based on local hillslope properties in this case.



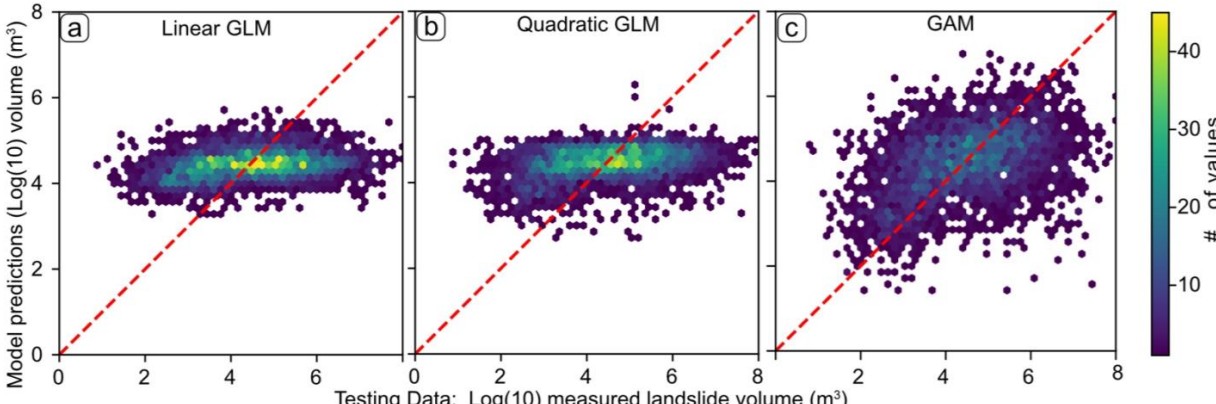

**Figure 6: Comparison of measured landslide volumes and model predictions which we ultimately deem unsuccessful. These are blind predictions of volume for the 20% of the inventory withheld from model development. a) shows the predictions from a general linear model limited to linear fits, b) shows the predictions from a general linear model predicted with quadratic fits, and c) shows the predictions from a general additive model. a) b) and c) show models trained with slope unit extracted local parameters, for radially extracted predictor results see Supp. Fig. S4.**


| Predictor variable extraction technique | GAM | GLM1 | GLM4 |
|---|---|---|---|
| Radial mean | 1.1237 | 0.9767 | 0.9546 |
| Slope unit | 0.9615 | 1.075 | 1.051 |

**Table 1: Statistical regression model results presented as the IMMSE (improved minimum mean squared error).**


## 4.2 Damability function for the Oregon Coast Range

Through our logistic regression modelling, we solved for the damability function fit to the landslide dam/non-dam inventory for the Oregon Coast Range (*Damability$_{OCR}$*). The function is visible as shading in Fig. 7, and in the form of Eq. 2.

$$Damability_{OCR} = \frac{1}{1+e^{-2.5937(\frac{log_{10}(V)}{log_{10}(2.338 \times W_V)} - 4.0168)}} \qquad (2)$$

where *V* is the landslide volume in m$^3$, and $W_V$ is the valley width in m. Probabilities of dam formation given valley width and landslide volume are shown as gradational shades of grey from < 10% to > 90% likelihood in Fig. 7.

To deterministically solve for landslide dam formation or non-formation volume, we adopt the 50% likelihood contour of the damability function, which can be expressed as a function of valley width as:

$$V_{50} = 0.004 \, W_V{}^{3.861} \qquad (3)$$

$V_{50}$ is also referred to as the minimum dam forming landslide volume because it matches the volume of slide, above which dams will most likely obstruct the valley.





Assuming a constant log-normal volume distribution across the study region (Fig. 7, left panel, see Section 3.3, 4.1), we compute damability, combining uncertainty in the damability function and range of expected landslide volumes resulting in Eq. 4 (Fig. 7 top panel):

$$Damability_{OCR-V} = 1 - \frac{1}{1+e^{-4.575(W_V - 1.745)}}$$ (4)

Equation 4 (*Damability_{OCR-V}*) includes both the logistic regression fit to the local landslide dam/non-dam inventory, and the lognormal volume distribution of local landslide dams. Damability values computed using Eq. 4, range from zero to one, and reflect the probability a
landslide forms a dam in a valley of a certain width, given the distribution of landslide volumes observed in the OCR and the uncertainty in the damability function.

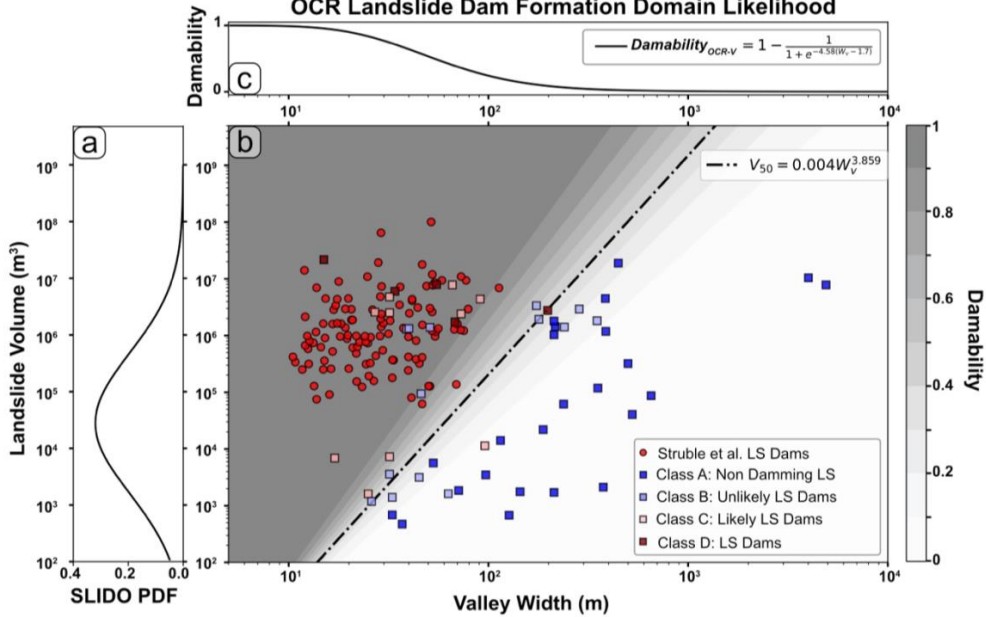

**Figure 7 Landslide damability functions. a) distribution of landslide volumes in the Oregon Coast Range. b) The same data as plotted in Fig. 3c shading from grey to white shows the probability of a width-volume pair being in the formation domain.**
**c) integrated likelihood of dam formation as a function of valley width (uncertainty in volume and damability function included)**

We used the dams mapped by Struble et al. (2021) (shown as yellow dots in Fig. 1) to explore the validity of this method. River stretches with mapped landslide dams have much
larger damability and dam susceptibility values than the general population of all river stretches in the study area (Fig. 8a). Most of the previously mapped landslide dams lie on river stretches with dam susceptibility values greater than 0.5 (78% of landslide dams, vs 28% of all river points) (Fig. 8a). We acknowledge that this line of model verification is somewhat circular, as the dams were used to calibrate the damability function that sets the dam susceptibility values,
and sometimes past dams can result in the narrowing of valleys where the river cuts through the dam deposit. We also randomly withheld 12.5% of the dam forming points and dam non-forming points, fit a new damability function, and checked the withheld points to the new function. The





new function does not deviate much from the fit to all data points and fits all the withheld datapoints (Supp. Fig. S6). We also note that the damability values are calculated with
algorithm-extracted valley width values rather than the map measured valley widths used in the calibration.

To further test the performance of the method and to identify a threshold to separate out river stretches with high dam-formation potential, we used a Receiver Operating Characteristic (ROC) curve. ROC curves compare positive results (river points with a dam) and negative results
(river points without a dam). However, we do not have many mapped river stretches without dams so we substituted the entire population as "negative results", which we find reasonable considering that there are <300 mapped dams compared to the 185 thousand calculated river points. Though not as ideal for area under the curve model verification (AUC=0.834) this simplification is ideal to find the appropriate threshold to use to separate river points with dams
from all river points. We found this threshold to be 0.35 by maximizing the "true positive rate" minus the "false positive rate." As a result, river stretches with landslide dam susceptibility values of 0.35 or greater should be considered river stretches with *dam potential*.

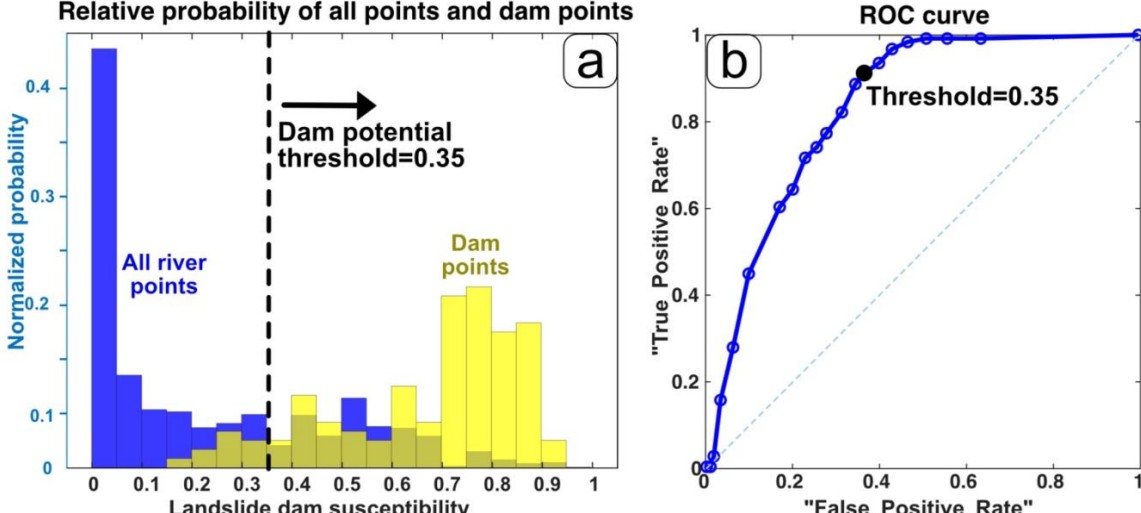

**Figure 8: a) Normalized histogram of landslide dam susceptibility values for all river points (blue), and points closest to**
**mapped dams (yellow). b) Receiver Operating Characteristic (ROC) curve for landslide dam susceptibility values comparing river stretches with mapped landslide dams and all river stretches.**

### 4.3 Damability, dam susceptibility, and lake volume

Across the study area, 51% of the calculated damability values are under 0.25, 19%
between 0.25 and 0.5, 20% between 0.5 and 0.75 and 10% greater than 0.75 (Supp. Fig. S5). Dam susceptibility values follow a similar distribution but are skewed lower, with 57% less than 0.25, 15% between 0.25 and 0.5, 18% between 0.5 and 0.75 and 10% greater than 0.75 (Fig. 9). Low values are prominent in large rivers, low elevation tributaries, and valleys draining to the east. High values are prominent in higher elevation rivers, most low drainage area tributaries,
and the core of the mountainous areas (Fig. 9)





Figure 9 shows that notable large areas of higher dam potential are found in the mountains around Tillamook Bay, inland Siletz River catchment, and in the headwaters of the Coos and Coquille rivers. In general, catchments draining east have significantly lower landslide dam susceptibility values than those draining west. In addition, tributaries and more mountainous river stretches tend toward the higher landslide dam susceptibility values, whereas coastal regions and the eastern edge of the study area record lower values.


To illustrate catchment scale patterns and differences in landslide dam susceptibility values across the OCR, we highlight two catchments, the Wilson and Alsea Rivers (Fig. 9 b and c). These drainage basins have similar drainage areas and both drain west, but they also represent the variation in landslide dam susceptibility values across the OCR, making them useful to compare. Within the Wilson River catchment, nearly all the tributaries contain high landslide dam susceptibility and much of the main river stems have moderate to high landslide dam susceptibility. In contrast, within the Alsea River catchment, only some of the tributaries reach high dam susceptibility, and much of the main stem has relatively low dam susceptibility.





**Figure 9: Landslide dam susceptibility estimates for the Oregon Coast Range. a) shown as averages over ~6 km across or ~30 km² area hexagons as well as for individual 100m long river stretches. b) Wilson and c) Alsea catchments. Mapped landslide dams shown as green diamonds.**





Using the relationship of Argentin et al. (2021), measured drainage areas, and estimated
minimum damming landslide volumes ($V_{50}$) from Eq. 3, we estimated the possible impounded
lake volumes at each river point. The lake volume estimates primarily correlate with river
drainage area (Fig. 10). Lake volume estimates for every river point span 11 orders of
magnitude, while the volume predictions at potential dam sites are significantly lower and peak
at closer to 1000 m$^3$(Fig. 10). Within the Wilson River basin over 69% of the river stretches are
above the potential dam threshold, and much of the main stem, with higher estimated lake
volumes ($>10^4$ m$^3$). Within the Alsea basin only 40% of the river points are above the dam

**Figure 10: Representation of Landslide dam potential across the study area. a) Drainage basins are colored along a greyscale by the percent of river stretches within the basin that have dam potential values above 0.35 the dam potential threshold. Only those potential dam river points are plotted as blue scale dots where the shade of blue is represented by the estimated dam lake volume that would form at that point. b) and c) depict the Wilson and Alsea catchments respectively. In insets river locations with dam potential values less than 0.3 are plotted as a black line.**



potential threshold. The Alsea River mainstem primarily does not have points above the potential dam threshold values (Fig. 10), meaning that most of the susceptible river stretches within the basin would hold relatively small lakes.


## 5 Discussion

### 5.1 The Damability function

#### 5.1.1 Damability function form

520       We present an approach to estimate landslide dam susceptibility using a logistic regression to find a probabilistic damability function. Past studies implementing damability functions, using datasets from Italy, (Tacconi Stefanelli et al., 2016), and the Cordillera Blanca in Peru (Tacconi Stefanelli et al., 2018), relied on manually placed lines separating formation, non-formation, and uncertain domains. The probabilistic approach to the damability function

regression that we present here, allows for a better characterization of the uncertainty of dam formation. Additionally, the probabilistic approach collapses the damability result down to one dam formation probability value, rather than one of three domain positions, which makes adding other useful metrics (i.e., landslide susceptibility, or estimated dam lake volume) more straightforward, and simplifies hazard visualizations.

530       The damability function regression methodology is flexible and can be applied to any dam/non-dam landslide inventory. The damability function for the OCR used in this study is found using the new morphology-based landslide dam/non-dam inventory we generated. While we have the best fit currently available, new input can affect the damability function slope and uncertainty. Additional landslides, especially those corresponding to underrepresented valley

widths or volumes, could alter the shape of the function (Supp. Fig. S7).

      The form of the damability function represents how efficient a slide of specific volume is at running out and damming a river valley. Studies have shown that, broadly, landslide runout scales as a function of landslide volume (Corominas, 1996; Pollock, 2020; Whittall et al., 2017). Pollock (2020) fit the largest dataset of landslide volumes and runout length data to find a

function relating the two, which if rearranged into the form of Eq. 3 has an exponent 2.87, and coefficient of 0.0018.  The line representing this function in log-log space has a slope between the Italian damability functions and the OCR damability function and is shifted towards larger lengths (Supp. Fig. S7). The shift towards larger lengths suggests that landslides of a given volume have a consistently larger runout length than the width of the valley that they can dam.

The fact that landslide volume to runout length is broadly consistent across landscapes, suggests that new dam/non-dam landslide inventories from other regions are likely to be reasonably fit by a damability function. However, further work is needed to assess if it will be possible to move beyond the need for local calibration of regional damability functions.

      In the two sites where we do have dam/non-dam inventories (Oregon and Italy) we find

that distinct damability functions fit the two areas (Supp. Fig. S7). The Oregon damability function has a lower Y intercept suggesting small slides can dam larger valley widths. However, due to the lower slope of the Italian damability function ($V_{50}$ exponent is 1.67 vs 3.8 for OCR), large landslides in Italy seem to dam larger valley widths than slides with a similar volume in the OCR. We speculate that small slides in Oregon may be dominated by long runout debris flows

while large slides are dominated by rotational failures with less of a flow style / long runout compared to large Italian slides. Alternatively, differences in damability functional forms may be



related to data gaps and outliers in the calibration slides. Local geology, geomorphology, and climate all likely control the form of the damability function, and regional differences in damability functions should be explored further with additional study sites and dam/non-dam
calibration landslides.

### 5.1.2 Landslide volume characterization

Using a damability function for susceptibility analysis requires study-area wide measurements of valley widths and landslide volume estimates. While we did not test various
methods for valley width estimation in depth (See Supp. Fig. S1 and S2), we did explore multiple methods for landslide volume estimation.

There are reasons to suspect that landslide volume is not constant across the study area. For example, within the Tyee formation landscapes dominated by small shallow landslides are characterized by steeper more planar slopes with high drainage densities and may have narrower
valley widths. Alternatively, landscapes with high densities of large deep seated landslides are characterized by scalloped hillslopes and wider (or more variable) valley widths (LaHusen and Grant, 2024). As such, the narrow valleys where damability values are high may only be exposed to small volume shallow landslides; while the wider valleys, where damability values are low, may be more likely to see large volume deep seated landslides. In this case, we may be
underestimating the damability for the wide valleys because they may be more likely to experience a large volume landslide than the rest of the study area.

This concern motivated our unsuccessful attempt to estimate variable characteristic landslide volumes across the landscape. We used measurable (at the regional scale) properties of the landscape to find a regression model to predict landslide volume (Fig. 6). We did not
satisfactorily predict the SLIDO landslide volumes. This is in contrast to the studies of Lombardo et al. (2021) and Moreno et al. (2022), who implemented versions of a generalized additive model (GAM) to predict the maximum landslide surface areas within a given slope unit.

Methodological differences may play a role in these conflicting results (See Supp. Fig. S4), though we suspect that the input landslide inventory used as the training dataset makes the
largest difference. Both Lombardo et al. (2021) and Moreno et al. (2022) used datasets from earthquake triggered landslides. Earthquake triggered landslides may have different characteristics, including failure style, than precipitation triggered landslides (Densmore and Hovius, 2000; Meunier et al., 2008). Landslides triggered by the same earthquake shaking, on the same day, have a limited spatial extent and may also exhibit similar properties on similar
slope types. The SLIDO inventory includes a wide range of failure styles, includes landslides which occurred at different times by different triggers on adjacent slopes, and was mapped by several different authors.

The volume of a landslide is controlled by many factors including: the height and width of the hillslope, the area of the unstable terrain, the distribution of frictional strength parameters,
the depth and shape of the failure plane, the triggering forces, and the landslide type. The complex interplay of these parameters makes estimating the volume of a possible future landslide on any given slope difficult. The interplay of these variables, may also mean that slide volumes are controlled by hyper local properties such as the variability in cohesion or groundwater recharge (Bellugi et al., 2021; Montgomery et al., 2009). In reality, a hillslope in
our study area could fail in a variety of ways, meaning that the landslide volume is not necessarily predictable by the variables we can measure at regional scales. The predictability of



earthquake triggered landslide sizes (only maximum area per slope unit) may be exceptional when it comes to landslide size predictions.

These results suggest that it is not straightforward to use hillslope properties or geometry as a proxy for landslide volume in landslide dam susceptibility analyses. McMeckin (2022) used relative relief as a proxy for landslide size and Wu et al. (2024) used delineated slope unit area as a proxy for landslide size. It is possible and intuitive that these proxies may match the largest possible landslides a hillside could produce, which may be the most important factor for landslide dam analyses. The benefits of slide size proxies, including the potential to eliminate the

need for rare, detailed landslide inventories, should motivate future research exploring controls on landslide volumes, especially in the context of landslide dam susceptibility analysis.

Instead of proxies, or a local regression model, we are forced to use a region wide empirical approach to landslide volume estimation based on the mapped landslides within the study area. Tacconi Stefanelli et al. (2020) also used mapped landslide inventories by

implementing a spatially determined power law exponent to estimate landslide volumes. However, patchy spatial coverage of the SLIDO landslide inventory, inconsistent lower volume threshold recording, and the lack of spatial coherence in landslide volumes in places where the inventory is complete make this less applicable than the overall log-normal volume distribution. Using a single distribution works well for dam susceptibility analysis in regions where landslide

inventories exist. In this study we must assume that landslide volume remains unpredictable, and proceed with susceptibility estimates dominated by valley width, which we can justify by the predominantly narrow valley widths at the mapped landslide dam deposits (Fig. 8a) (Struble et al., 2021).


### 5.1.3 Future applications

We expect that the damability function methodology – as it is currently presented – will
be applicable to other regions, especially neighbouring areas of the Pacific Northwest of the US, although it will always be limited by available data sources. Valley width estimations using our algorithm are only applicable in locations with a 10 m resolution or better DEM. Detailed landslide inventories are required to characterize the landslide volume distribution and landslide dam/non-dam inventories are needed to refine the local damability function. While the

methodology of the landslide mapper may vary between studies, because this method only uses the inventory to define a lognormal volume distribution it is insulated from variations in inventory quality and completeness. Lastly, landslide susceptibility information is also required and often defined using a combination of landslide inventories and geomorphic measurements. These requirements show the importance of detailed landslide deposit inventory mapping and

suggest that including a valley connection term and dam/non-dam value would be a helpful addition to the list of data attached to landslides in inventories. The damability approach presented here can be modified to use different methods for measuring valley width and estimating landslide volume, and it can be recalibrated with new data from other regions. Large regional studies will have large uncertainties because various landscapes with different

lithologies, glacial and tectonic histories, fluvial erosion patterns, and climatic controls will be included. However, more large-scale studies are necessary to quantify landslide dam hazards and can be completed using the damability function approach.





## 5.2 What controls regions of higher landslide dam susceptibility?

Due to the nature of our approach, relative differences of dam susceptibility values across
the study area are almost entirely driven by differences in valley width. The damability function
and landslide volume characterization are constant across the study area. The landslide
susceptibility factors only have a small effect on spatial trends, as most locations with high
damability also have high landslide susceptibility.

In general, valley width increases with increasing drainage area, resulting in lower
landslide dam susceptibility values at high drainage areas and at lower elevation and coastal
regions (Fig. 11a). While local channel widths are modulated by contemporary fluvial properties
including river slope, discharge, bank strength, and bed material (DiBiase and Whipple, 2011;
Dunne and Jerolmack, 2020; Leopold and Maddock, 1953), valley widths reflect erosional and
aggregational histories making them more difficult to predict (May et al., 2013). In much of the
OCR, valley width is found to vary with the drainage area raised to a power (Supp. Fig. S2)
(May et al., 2013). However, in drainage basins where deep seated landslide morphology is
common, the valley width vs. drainage area relationship is more scattered, though the general
trend persists (May et al., 2013). Across our study area, we observe that river headwaters, where
drainage areas and measured valley widths are low, are more susceptible to landslide dam
formation than downstream in high drainage area rivers.

Rivers that flow through more mountainous terrain have higher landslide dam
susceptibility. Clear correlations are visible in the relationship between dam susceptibility values
and area averaged slope and relief (Fig. 11b,c). In general, higher slope and relief correspond
with river headwaters, but mainstem rivers flowing through mountains also have higher dam
susceptibility values. In mountainous reaches, hillslope processes such as landsliding, valley wall
armouring, and debris flow sediment input control valley width (Grant and Swanson, 1995;
Shobe et al., 2021). Within the OCR, valley widths are generally depressed in basins correlated
to landsliding, though it depends on the stream position relative to the landslide deposit (May et
al., 2013). East of the study area, on the west side of the Oregon Cascades, mountain valley
morphology was shown to be strongly controlled by bedrock exposure and hillslope and tributary
processes, including landsliding (Grant and Swanson, 1995). Hillslope sediment delivery is
proposed to control valley width (Tofelde et al., 2022), as found for terrace edged alluvial
channels. Landsliding, bedrock exposure, and hillslope sediment supply all are expected to
increase with increasing slope/relief, and possibly uplift, which may keep mountainous valleys
narrower (e.g., Heimsath et al., 2012; Larsen and Montgomery, 2012; Roering et al., 1999).
Indeed, the locations in the OCR where we observe the highest dam susceptibility (Fig. 9)
spatially correspond with the highest uplift and erosion rates inferred from hilltop curvature
(Struble et al., 2024).






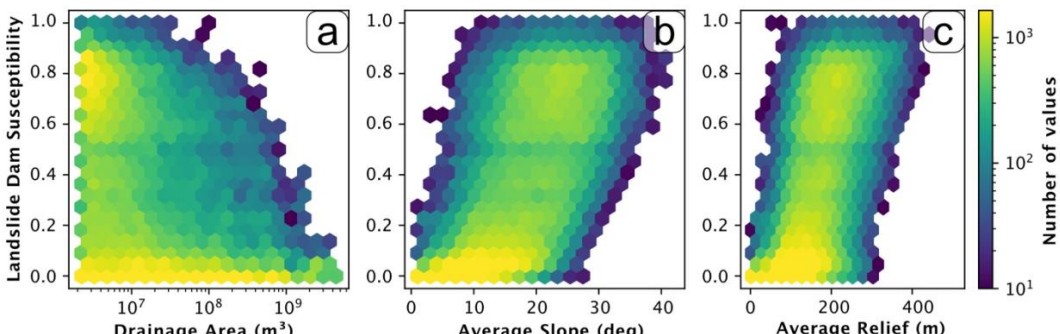

**Figure 11: Trends of the landslide dam susceptibility values, plotted with a) river drainage area, b) average slope (mean value sampled from a 500m radius around the river point), and c) relative relief (also mean value sampled from a 500m radius around the river point). Colour scale for all plots represents the number of river stretches with values corresponding to the hexbin.**

690        Lithology also likely plays an important role in setting landscape morphology and, consequently, dam susceptibility. Figure 12 demonstrates a correspondence between volcanic rock and higher relief (and landslide dam susceptibility) within the northern section of the study area, most notably exemplified by the Tillamook Volcanics around the Wilson River basin. The high relief areas in the south of the study area do not always correlate with a separate rock type,

though they may match inter-rock type variations in grain size and possibly erodibility(LaHusen and Grant, 2024). This is consistent with the findings of Schanz and Montgomery (2016) who compared basins just north of the OCR dominated by igneous bedrock, with the areas of the Nehelam river (Northern OCR Fig. 9) dominated by friable sedimentary rock. They found valley widths to be 2-3 times wider in the sedimentary basins and proposed that this was due to the ease

of lateral erosion in those easily weathered rock types. In the Himalaya uplift rate, rather than rock type was found to be the largest control on valley width (Clubb et al., 2023). However, uplift rates within the OCR span only a narrow range, which may allow the variations in rock type to play a key role in setting valley width, similar to their influence on hillslope form (Struble et al., 2024). For instance, more resistant lithology may inhibit lateral erosion in

channels, thus promoting narrow rivers (e.g., Li et al., 2023) and also support greater landscape relief (e.g., Neely et al., 2019).

        Volcanic rocks may host narrower rivers; however, they must also host large landslides to lead to formation of landslide dams. While it may be intuitive to expect that lithology will control landslide size and that stronger rocks may form smaller landslides, in this study we find

no evidence that landslides in volcanic rocks are smaller than those in other lithologies. Lithology was included in our attempt to predict local landslide volumes with multivariate regression (Fig. 6). There are no large differences in the volume distributions of landslides based on lithology (Fig. S8). In fact, the lognormal mean for volcanic rocks is slightly higher than the mean for the marine sedimentary rocks that make up most of the study area 4.6 to 4.4

respectively. It is possible that the landslide frequency varies across lithologies, however tackling this question is beyond the scope of this work (and capabilities of the incomplete inventories available). Our analysis indirectly accounts for relative frequency by using the state-wide landslide susceptibility map which incorporates lithology (Burns et al., 2016). This map and other susceptibility maps that incorporate lithology do not show a decrease in landslide

susceptibility in regions of volcanic rocks (Burns et al., 2016; Sharifi-Mood et al., 2017). Further





research could inform how geology controls possible dam forming landslides. For now, we conclude that (at a regional scale) lithology is not the primary control of landslide character, but does likely act as an important control on valley widths, and consequently landslide dam susceptibility.

725         The distribution of landslide dam susceptibilities found in this study are generally similar to other landslide dam susceptibility studies. Studies in New Zealand's Southern Alps (McMeckin, 2022), Italy's Arno Basin (Tacconi Stefanelli et al., 2020), and Central Asia (Tacconi Stefanelli et al., 2023), generally predict a much higher percentage of river stretches with low landslide dam hazard than with intermediate or high hazard. McMeckin (2022) found

roughly 28% of the points in the catchments they studied at risk, which is comparable to the 36% of points above dam potential of 0.35 in our study. In both the Arno Basin and Central Asia, around 30 to 40 percent of the rivers were associated with moderate, high, and very high damming susceptibility (Tacconi Stefanelli et al., 2020, 2023). All studies show a clear relationship between mountainous areas with high relief and areas with higher landslide dam

susceptibility.

        We suggest that landslide dams are more likely to occur close to river headwaters, along stretches of river that cut through higher topography, and through resistant rock types. The location of the steep terrain and narrow valleys is likely controlled by lithology in the OCR but may have other controls (such as uplift rate) globally.




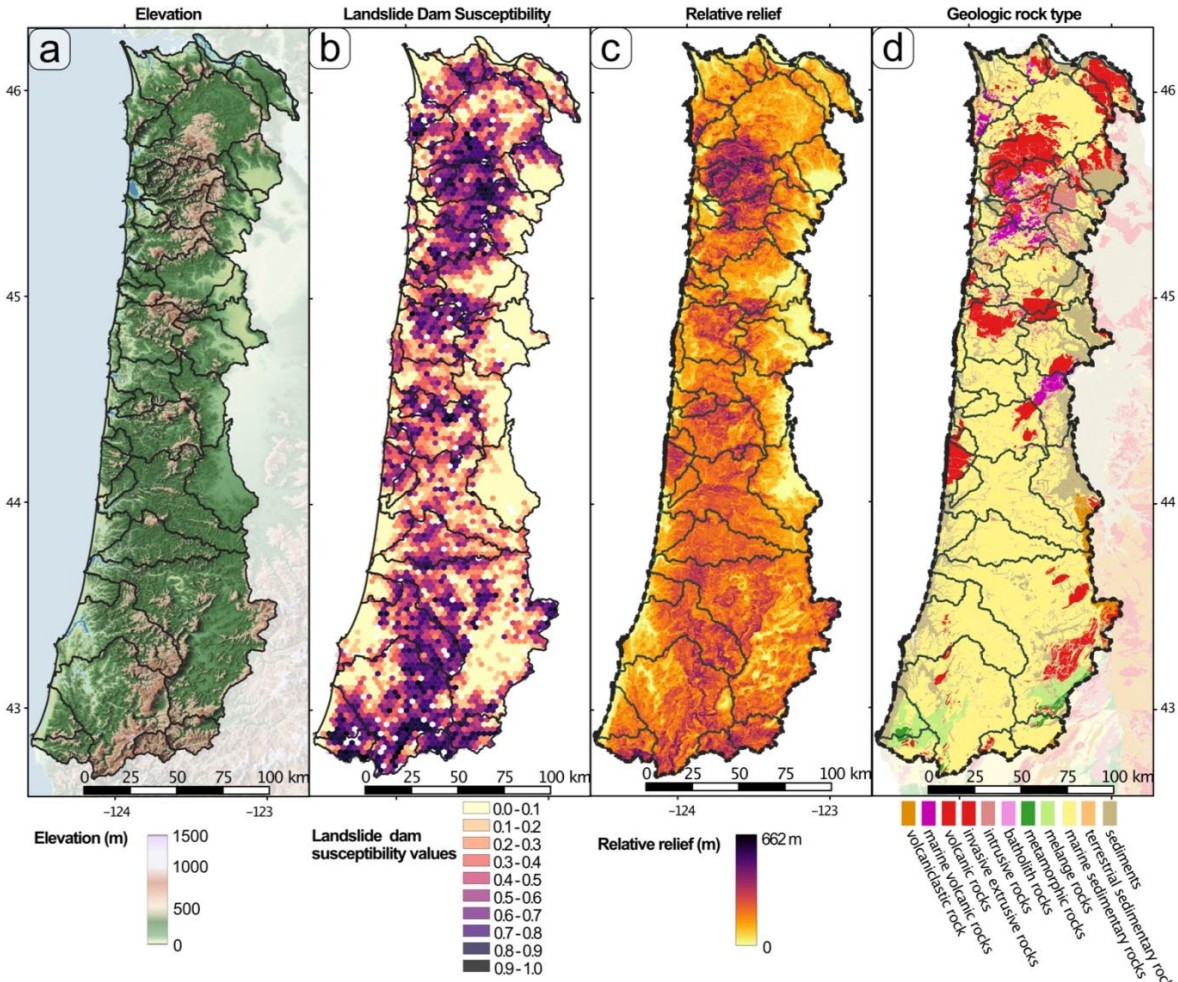

**Figure 12: a) Elevation b) Dam potential, c) relative relief, and d) rock type for the study area presented side by side to showcase correlations. Dam potential values are averaged over each ~3km across or 7.5km² area diameter hex bin. Relative relief values are calculated over a 500 m moving window on the USGS 1/3 arc second (10m) 3DEP program DEM (US. Geological Survey, 2023). Rock type values are from the Oregon Geologic Data Compilation (OGDC-6) by DOGAMI.**

## 5.3 Landslide dam hazards in the Oregon Coast Range

The potential for landslide dam formation is present and widespread across the OCR. Roughly one third of the river points have dam susceptibility values greater than 0.35, the threshold placed to define potential dam sites (visible points in Fig. 10). These results are consistent with the large number of landslide dam deposits (>200) observed throughout the study area (Struble et al., 2021) and suggest that flooding associated with landslide impounded riverways is a hazard worthy of consideration in community planning.

Fortunately, most of the river stretches with high potential for dam formation would likely hold small lakes ( $<10^4$ m³) (Fig. 13). Rivers with high landslide susceptibility values can be dammed by smaller landslides. Our lake estimate procedure adjusts the slide volume based on the minimum damming slide volume ($V^{50}$) for the river stretch, therefore we predict smaller lake





volumes for narrower river valleys. We consider our predicted lake volumes (yellow line in Fig.
11) low when compared to a dataset of historic landslide dam outburst floods (Costa and
Schuster, 1991), where lake volumes of 1 million cubic meters are the minimum recorded lake
volumes for catastrophic outburst floods (black dots in Fig. 13a). Smaller landslide dam lakes are
still capable of causing damage and may still require a swift mitigation response, (Costa and
Schuster, 1991), however the 1 million cubic meters mark provides a good approximation of
extreme impact potential.

Landslide dam disasters are usually triggered by exceptionally large landslides (Costa
and Schuster, 1991). Two examples of past landslide dams which occurred in the Pacific
Northwest, the Oso landslide and the Bonneville landslide dam, dammed rivers in wide valleys
which would have low damability values ~0.1 and ~0.01 respectively (Iverson et al., 2015;
Pierson et al., 2016). These were rare and large landslides, lying at the 98th percentile (Oso) and
99.9th percentile (Bonneville) of the SLIDO volume distribution. Large landslides like these are
also capable of damming relatively larger rivers with lower gradients that can lead to rapid lake
filling and extensive upstream impacts.

If we estimate landslide dam lake formations with a worst-case assumption of a landslide
with a volume corresponding to the 95th percentile of the SLIDO inventory ($3x10^6$ m$^3$) the
magnitude and location of dams with the greatest potential impacts shifts dramatically. This
worst-case landslide would likely (damability>0.5) dam 66% of the river stretches, and possibly
(damability>0.1) dam 88%. The estimated lake volumes at those likely dammed sites
dramatically increase as well (red dashed line Fig. 13a). In a few rivers the 95th percentile
landslide would likely dam the river and could impound a lake exceeding the 1 million cubic
meters associated with historic catastrophic outburst floods (Fig. 13a). The corresponding worst-
case lakes (Fig. 13b) are primarily in the largest rivers in the study area and are not correlated to
the catchments with the highest landslide dam susceptibility values.  The mismatch between the





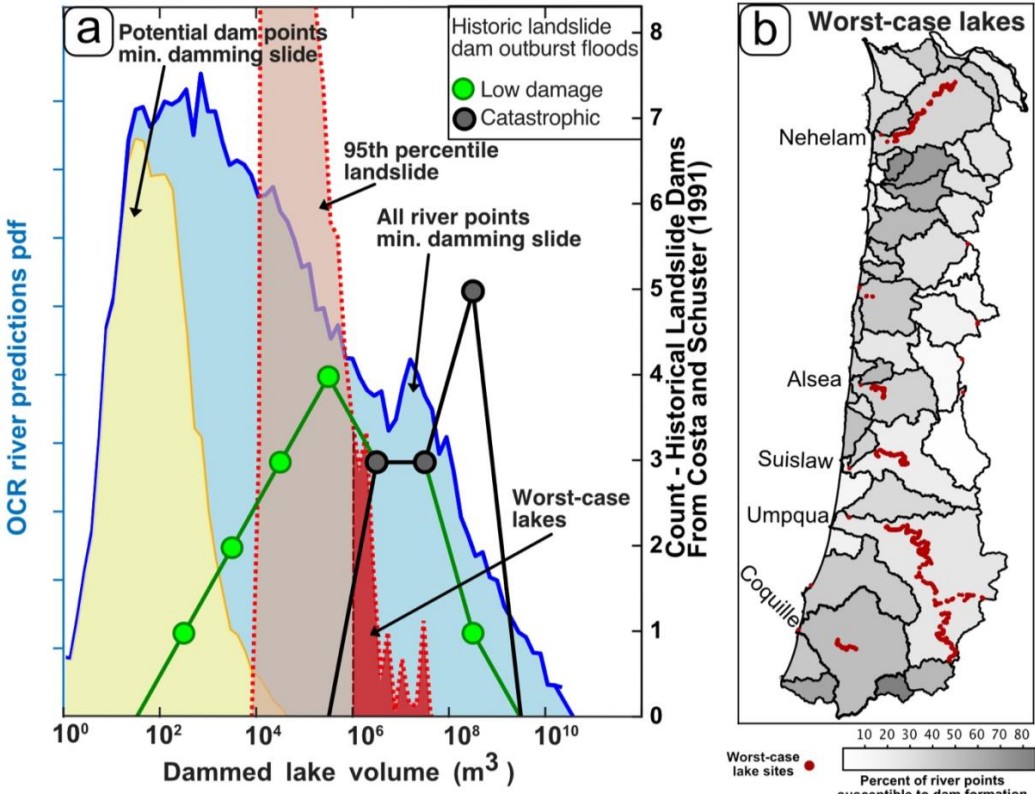

Figure 13: a) Comparison of the volumes of historic landslide dam lakes with predicted lake volumes for landslide dams at OCR river points. Green and black dots represent a subset of the data presented in Costa and Schuster (1991) where dam lake volumes are recorded and outburst flood effects are noted, green dots caused little to low damage, and black dots caused catastrophic damage. Blue line represents dam volumes predicted for every river point, calculated with the minimum damming landslide volume ($V_{50}$ EQ. 3). The yellow line represents the same dataset as the blue line, but limited to only potential dam sites where landslide dam susceptibility is greater than 0.35 (Plotted in Fig. 8). Light red dashed line represents lake volumes estimated using the 95th percentile volume ($3 \times 10^6$ m$^3$) landslide, and limited to points where that large slide leads to damability values > 0.5, and dark red shaded area represents the wort-case lakes where the 95th percentile landslide might impound a lake greater than 1 million cubic meters. b) maps the locations of the worst case lakes in a) over the catchments shaded by percent of river points that are above the dam potential threshold, same as Fig. 10.

rivers where landslide dams may be the most likely to form, and those where the impacts could
be the greatest, presents a mitigation planning challenge worthy of further study.

An obvious next step would be to improve on this analysis through more detailed
assessments within the most hazardous catchments. This could bypass some of the
simplifications made to take advantage of regional datasets. Local lake volume estimates could
be improved using local topography rather than a scaling relationship not calibrated on fluvial
valleys (Argentin et al., 2021; Hergarten et al., 2023). Local evaluations may also be improved
by valley specific geotechnical investigations to characterize local landslide susceptibilities as
well as constrain landslide volumes and possible runout distances. For example, landslide
patterns within the study area are known to be controlled by sub geologic unit lithologic
properties (Roering et al., 2005; LaHusen and Grant, 2024). That level of detail is beyond the
scope of this regional analysis

Ultimately the risks to populations posed by landslide dams are controlled by the location
of the landslide dam formation, the size and stability of the dam, the size of the impounded lake,




and the exposure of upstream and downstream people and infrastructure. The scope of this study is to identify the locations where landslide dams may be most likely to occur, and place initial
estimates on the magnitude of the impacts. Our results can help mitigation planning, prioritize valleys for localized detailed analysis, and order initial inventory scouting in the case of a widespread landslide triggering event. Future work could identify the most exposed communities, investigate the landslide dam susceptibility upstream of those communities, and simulate outburst floods within those valleys. How landslide dam hazard analysis work is
continued and presented should represent a cross section of what is scientifically possible and be informed by what products end users need (e.g., Barnhart et al., 2023).

**6 Conclusions**

In this study, we refined a methodology used to estimate landslide dam susceptibility at a
regional scale in the Oregon Coast Range (OCR). As part of this workflow, we present a new methodology for fitting a dam/non-dam inventory to derive a probabilistic damability function which can be used to predict landslide dam likelihood given landslide volume and valley width. We show that landslide volume is not always predictable based on local geomorphic and geologic factors. Our case study demonstrates that a log-normal distribution of landslide volumes
and our valley width measurement algorithm can be successfully used to assess landslide dam susceptibility using a damability function. We also demonstrate how damability function derived damability values can be used in conjunction with landslide susceptibility or impounded lake volume estimates to better characterize and visualize landslide dam formation likelihoods and magnitudes. The damability function approach is flexible to the input of new data and can be
recalibrated by existing or new inventories for use in other regions globally.

When we apply this new methodology to the OCR, we find that dam susceptibility correlates with drainage area, mountainous terrain, and lithology. High landslide dam susceptibility is widespread across the OCR, where roughly one-third of the river stretches have landslide dam susceptibility values above the dam potential threshold value (0.35). In most of
these susceptible river stretches, we estimate that landslide dams would impound relatively small lakes that are generally less dangerous. However, exceptionally large landslides could dam most of the OCR rivers and impound lakes large enough for catastrophic outburst flooding in some rivers. These results show that landslide dam hazards could be significant in the Pacific Northwest. If a large, widespread landslide triggering event, such as a subduction zone
earthquake, were to occur, we expect that landslide dams may form in the OCR, and the chances of a landslide dam leading to a catastrophic outburst flood are non-negligible.


**Code Availability**

The code used to measure valley widths is available at https://github.com/PMonroeMorgan/valleywidth.



**Data Availability**

We provide a supplemental data files including the dam/non-dam inventory used to fit the damability function, and the analysis results at all river points. Much of the data used in this study is publicly available and provided by the Oregon Department of Geology and Mineral Industries (DOGAMI) including: the Lidar data (https://www.oregon.gov/dogami/lidar/Pages/index.aspx), geologic map data (https://www.oregon.gov/dogami/geologicmap/Pages/index.aspx), and landslide inventories (https://www.oregon.gov/dogami/slido/Pages/data.aspx).


**Author Contributions**


AD conceptualized the research project. PM performed the investigation, developed the methodology, and wrote subsequent software. PM and ag performed the analysis. PM, ag, WS, SL, and AD contributed to interpretation of results. PM wrote the manuscript draft. ag, WS, SL, and AD reviewed and edited the manuscript

**Competing Interests**


The authors declare that they have no conflict of interest.

**Disclaimer**

Any use of trade, firm, or product names is for descriptive purposes only and does not imply endorsement by the U.S. Government.

**Acknowledgements**


The authors gratefully acknowledge funding support from National Science Foundation (NSF) Grant Number 2103713 to Duvall. We thank N. Calhoun and B. Burns for comments and recommendations in the early stages of this work. The manuscript was improved by comprehensive reviews by internal U.S. Geological Survey researchers A. Dunham and S. Ahrendt. Any use of trade, firm, or product names is for descriptive purposes only and does not imply endorsement by the U.S. Government.


**Financial Support**

National Science Foundation grant ICER-2103713.

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
