# Peer review of "The damability function: A probabilistic approach to regional landslide dam susceptibility analysis applied to the Oregon Coast Range, USA"

_EGUsphere, 2025_

## Author Comment (AC1)

**Reviewer number 1**

Format of this document:

Comment text

Response text

"modified text snippet"

**Dear Authors,**

In this study you are concerned with estimating how likely it is to have a river dammed by a landslide in the Oregon Coast Range, United States. You draw on data of nearly 20,000 previously mapped landslides and several hundreds of landslide dams, and present a logistic regression that outputs the probability of recovering a (known) landslide dam as the target variable. You consider landslide volume and valley width as predictors and thus obtain a quantitative estimate that you term "damability". You find that the potential for damming is likely higher in steeper headwaters than in larger rivers with wider valleys. You cast your results in several maps and also estimate the water volumes likely to be impounded in damming scenarios. This type of probabilistic treatment is long overdue, and I am happy to see that you tackled it.

Overall, your study surely fits the scope of the journal and may well be of interest to both scientists and practitioners dealing with landslides, rivers, or mountain hazards in general. You present your study well, albeit in a bit longish format that could benefit from some shortening (without losing any information). I would like to see this published eventually, but I also believe that you may need to consider a number of items beforehand:

■ Thank you for this complete and accurate description of our study. We removed some sections from the manuscript while also addressing the general and specific comments.

--- General Comments ---

The abstract is easy to understand, but could also reveal some more details or benefits concerning the predictions from your damability function.

■ We have added some language to better highlight the results from the Damability function section of our analysis. "We validate and apply our approach to the Oregon Coast Range, USA and find that 36% of river stretches exceed a dam potential threshold"

The introduction (section 1) could explore a bit more any previous work that tried to figure out where landslide dams occur most likely. Consider reviewing in more detail some attempts at prediction. You might also want to summarize briefly what the many inventory studies tell us about the relative catchment position of landslide dams.

■ We have significantly expanded the paragraph on previous work to two longer paragraphs, moving some information from section 3.1. The introduction now includes the following two paragraphs:

"The review of Fan et al., (2020) nicely summarizes and tests several metrics used to predict landslide dam formation. These metrics include: the 'annual constriction ratio' (ACR) which concerns the ratio between landslide velocity and valley width (Swanson et al., 1986), the 'Dimensionless Morpho-Invasion Index' (DMI) which is a more complex formulation of the ACR incorporating more physical parameters of the landslide like velocity, density, volume, and hydraulic level (Dal Sasso et al., 2014), the 'Dimensionless Constriction Index' (DCI) can also be used for stability estimates and includes parameters of the landslide like geometry, velocity, and grain size, as well as the width of the valley (Ermini and Casagli, 2003), and the 'morphological obstruction index' which concerns the ratio between log valley width and landslide volume (Tacconi Stefanelli et al., 2016). The ACR, DMI, and DCI require estimates of landslide velocity, which is difficult to measure for a known landslide, and even more difficult to estimate for ancient or future landslides across a landscape. The MOI (Tacconi Stefanelli et al., 2016) is the simplest and the easiest to implement at landscape scales for situations where the properties of a future landslide must be inferred.

Three methodologies have been used to create susceptibility maps, all following roughly the format of the MOI by taking into account landslide volume and valley or channel width. Tacconi Stefanelli et al., (2020) introduced a semi automated workflow to estimate susceptibility across the Arno river basin in Italy and used the landform classification to measure valley width, and a spatially variable power law fit to a large landslide inventory to estimate landslide sizes. McMeckin (2022) used a somewhat similar technique in the West Coast region of New Zealand and used relative relief as a proxy for landslide size. Wu et al. (2024) developed globally applicable methodology that uses a global river dataset of river width and slope unit area as a proxy for landslide size. However, this method relies on global river datasets that do not include rivers with sufficiently small drainage areas to capture and calibrate with landslide dams the OCR, namely those in Struble et al., (2021)."

The outline of the study area (section 2) is a bit short and may benefit from some more pointers to studies that readers can chase up. The landslide (dam) inventories do need more detail here, though, as the form part of your training data set. Please provide some more background and insights from these inventories.

■ We expanded this section and included more details about the inventories that we use in this study. In some cases, moving text from elsewhere. We agree that this is a natural place to find information about the datasets we are using that record information within the study area. The following two paragraphs include expanded information on the inventories.

"The SLIDO database is a compilation of landslides that appear on published maps. Many of these landslides were mapped on LIDAR (light detection and ranging) generated digital elevation models follow a protocol defined by "Special Paper 42" (Burns and Madin, 2002) which require the mapper to measure, estimate, or calculate additional parameters of the landslide, such as volume. We use landslides that were mapped using the special paper 42 protocol found in SLIDO version 4.5 to estimate landslide volumes in this study.

Landslide dams have been documented in the area by Struble et al., (2021), who published an inventory of 238 landslide dams (yellow dots in Fig. 1). These dams were primarily caused by deep seated translational or rotational slides. They found that preserved landslide dams are overrepresented in drainage areas of ~1.5 to 13 km² and valley widths of ~25-80 m. Despite the proximity of the Cascadia Subduction Zone, few dated landslide deposits or landslide dams can be correlated to earthquakes (Grant, Struble, & LaHusen, 2022; Struble et al., 2021; LaHusen et al., 2020). We use this landslide dam inventory (with additions) to fit the damability function for the OCR."

The methods section (3) could also be a bit more revealing, especially about the landslide data and their constraints:

section (3): What is the estimated error on the landslide volumes and were they all derived consistently? A potentially confounding issue is that you use two independent, and differing data sets. One has a couple of hundred landslide dams, whereas the other features more than 19,000 landslides. This choice may undermine some assumptions of likelihood-based model fits.

■ The landslide volume estimates are provided as part of the DOGAMI provided SLIDO landslide database. No errors for landslide volumes are presented in the database. This is a good point about differences in the inventories that we use. We address this comment on the variety of input datasets later in this response (see discussion of the logisitic fits.) We also have added a few sentences to the background section and methods section to help clarify the distinctions between the two landslide datasets we use and how we use them differently.

Section (3): Your general damability model (Equation 1) has an awkward specification, and uses the ratio of landslide volume to valley width, whereas you implicitly made the case in Fig. 3b/c that both these should be independent. Equation 1, however, specifies a regression that solely models the interaction effects between landslide volume and the inverse of valley width (plus an estimated off-set), hence some metric of "valley narrowness". Maybe emphasize your reasoning behind this more candidly.

■ We agree that the Damability model function has a specific form related to the ratio of the log transformed valley width and landslide volume. We impose this form to match previous research, namely the work of Tacconi Stefanelli et al., 2015. Who used this framework to predict landslide dam formation from a larger input dataset gathered from historical records. We tested the logistic fit using an additive form of V and Wv rather than the ratio, and found a similar fit. The plot is added below this comment. We feel that a full exploration of the choice of functional fit, (including the plot provided below) is beyond the scope of this manuscript but could be explored in future work that addresses several landslide dam inventories.

However, we have added text to explain why we use this functional form. "While alternative models (e.g., linear combinations of volume and width) perform similarly to our preferred model (Eq 2), we selected this model for both its good performance and to build on previous work where MOI has been found to be a reliable estimator of landslide dam formation. "

Section (3): Despite reading and trying to reproduce this section several times, I found it confusing, partly because (i) log-transformations occurred without consistent mention; (ii) the landslide volumetric distribution is included in a model designed to predict damming (instead of volumes); (iii) the model parameters eluded any description or interpretation. Why not simply use a logistic regression with landslide volume and valley width as additive predictors? This would be compatible with what you show in Figs. 3 and 7, while keeping the three fitting parameters, though in a much more accessible way (i.e. as marginal effects of landslide volume and valley width, if properly standardized).

- (i) We have added in some clarifying statements about where log transformations occur, including in "(all logarithms are base 10)" "Log base 10 transformed values of valley width and landslide volume were used to better model landslides and landscapes over several orders of magnitude in scale with a consistent relationship." We believe that the inconsistent mention refereed to is in reference to an equation where there is no log transformation intentionally, and we have added clarity to the equation descriptor "Where only valley width (Wv, meters) is a required input"
- (ii) We have added in section 4.2.1 to better clarify how the volumetric distribution is used in our workflow

**"4.2.1 Landslide dam formation likelihood**

To estimate the potential for future landslide dam formation given some unknown landslide volume and uncertainty in the damability function, we combine valley width measurements, the empirical distribution of landslide volumes (4.1), and Eq 3. For all valley widths, landslide dam formation likelihood (Fig. 7c) was computed as the sum of expected damability (Fig. 7b) for the entire lognormal distribution of expected landslide volumes (Fig. 7a). For ease of use, we then fit the landslide dam formation likelihood values as a function of valley width as..."

(iii) Please see our response to a similar question made in the previous comment. We appreciate the reviewer pointing out which parts of the methodology are less easy to follow. As we strive for clarity we have added more details throughout the methods section.

Section (3): For some reason, I could not arrive at some of your parameter estimates; perhaps I must have mistyped or overlooked some critical information (see below). Nonetheless, please describe your model setup as clear and rigorous as possible, so that your readers can reproduce it. Section 3.3.2 seems like a candidate section to drop entirely. If you decide to keep it, you should describe the underlying models in due detail.

- We especially appreciate this comment provided by the reviewer which demonstrates their attention to detail in reviewing this manuscript! To address a lack of clarity on how we reached certain parameter values in EQs 2-4, we have added in two equations to make the equation workflow easier to follow, and have added in section 4.2.1 (partially quoted a previous comment) to better explain the derivation of equation 3 (now equation 6).
- We acknowledge that since section 3.3.2 is not used in our final workflow that it could be cut from the manuscript. We have trimmed the results figures from the manuscript to reflect this. However, we feel that this workflow is well motivated, and that null results are important to present for future scientists approaching similar problems. Therefore, we opt to keep section 3.3.2 in the manuscript.

The results section (4) may also need some attention.

Section (4): The derivation of Equation (3) could be more explicit, the parameter estimates re-checked, and their errors propagated.

■ We have added in section 4.2.1 (partially quoted a previous comment) to better explain the derivation of equation 3 (now equation 6).

Section (4): In Equation 4 you express damability as a function of valley width alone, as opposed to Equation 3, where you used the ratio of landslide volume over valley width.

The simpler Equation 4 ignores the uncertainties of the log-normal fit to the landslide volumes: you boil down the number of parameters at seemingly no cost. You could argue equally well that the distribution of valley widths matters also, but you bring this up in the model validation later on. Please be consistent.

■ We believe that this comment demonstrates the need for an additional supplemental section on the derivations of Equations 2-4, and the lack of clarity we provided in the original manuscript. We have added in section 4.2.1 to better clarify the derivation of equation 4 (now Eq. 6). Equation 4 does not ignore the uncertainties in the landslide volumes. The uncertainty of the landslide volume, is held within the parameters (standard deviation) of the normal function fit to the log landslide volumes. And because the landslide dam likelihood is integrated across this distribution the uncertainty does contribute to equation 6.

Section (4): Again, I am puzzled why you discarded the logistic regression with landslide volume and valley width as additive predictors, thus implicitly accounting for the distributions in both (with their interaction arising naturally from the log10-transform). You could then still use the "SLIDO PDF" in Fig. 7 (and the distribution of valley widths in your study area) to estimate predictions.

■ We chose the set functional form for the fit based on the methods used to estimate a fit by Tacconi Stefanelli et al., (2015). Please see our previous comment where we describe why we made this choice.

Section (4): In this context, you might want to explain why the decision boundary in Fig. 7 becomes more precise with smaller landslides and narrower valleys: the transition between high and low damability estimates is more stepped for this configuration than for larger landslides in wider valleys. This model outcome is counterintuitive as your "SLIDO PDF" shows that your data density is quite low in this case. The decision boundary should be most distinct where data density is highest.

■ This is an interesting observation. Note that only the dam/no dam points that are plotted in the graph area are used to derive this function, the SLIDO pdf dataset is not used in this derivation, so does not influence the fit. We believe that the distance between points with known dam formation and non formation is a key driver of the uncertainty window shown. We have added discussion of this topic to the discussion section. "The width of the decision boundary (uncertain areas where damability values are between 0 and 1), may also be related to data gaps. It is narrower for low volumes and widths than larger volumes. Although this pattern is less pronounced for our fit to the Italian inventory (Fig. S7-b), a similar trend exists and both inventories have few small (< 105 m³) dam forming landslides"

Section (4): The part about estimating model performance (Fig. 8) brings in information more relevant to the methods (such as sample size, balance/imbalance), and again raises points better suited to the discussion.

■ We chose to put this section in the results because we felt that it explained the dam potential threshold result, as well as the result that the function is validated. But we felt the need to repeat some information (regarding sample sizes) from the methods (study area background) and also to keep some traditionally discussion information from crowding the discussion section.

The discussion (section 5) could be slimmer and more to the point.

Section (5): Besides trimming several wordy or overly repetitive statements, I suggest you transfer some major chunks of text to the study area and methods sections to fill in some of the gaps there (see specific suggestions below).

We agree that some of the study area information listed below should be presented in the study area section. However, for clarity we will repeat the necessary information here to make the points we are discussing.

Section (5): Another obvious asset would be to discuss in more detail the model performance and alternative formulations. For example, what other extensions to your model could you think of other than adding more predictors? You duly mention that only a fraction (of the order of a few percent) of the known landslides formed dams. In terms of logistic regression, this can be a problem, as the method assumes data that are roughly balanced across both classes (dam/no-dam). You can sample the larger data set as you did, but need to demonstrate that this sample is representative. Bootstrapping is one possible method, while rare-event logistic regression is another.

- We spend 5.1.1 discussing the form of the damability function. We have also added the following text to the methods section to discuss functional form "While alternative models (e.g., linear combinations of volume and width) perform similarly to our preferred model (Eq 2), we selected this model for both its good performance and to build on previous work where MOI has been found to be a reliable estimator of landslide dam formation."
- We have added detail through the expanded methods section to clarify that we only use the landslide dam/non-dam inventory which is roughly balanced between null and positive results in the logistic regression. Also data used in the logistic regression represent landslide-valley interactions, while not all landslides in the SLIDO inventory fit this subset, making it difficult to demonstrate representation.

Section (5): Why not use landslide runout instead of volume? Some aspects regarding your data might also need reflection: how many of the SLIDO landslides were catastrophic, how many were (or are) slow-moving such that, even if they have a large volume, they might not necessarily warrant river damming?

■ This is a good question. We agree that landslide type/style matters, and we discuss this in section 5.2. In many ways landslide runout better reflects the ability of a landslide to cross a river than the volume of the landslide, especially in the case of a large volume low displacement landslide. Other published metrics that are more physically derived do use landslide runout. However, estimating the runout of future landslides at a regional scale has not been attempted (and requires many assumptions) while landslide volume statistical fits and predictions have. This could be an avenue of future research.

The conclusions give a good summary, but could feature some more quantitative detail. I am not fully sure whether all your statements here are fully supported by the data, though (see below).

Please see responses to the line specific suggestions below.

--- Specific Suggestions ---

5: Typo in "alex".

■ Not a typo, a name preference not to capitalize

15: "landslide dam formation susceptibility" – Please avoid four nouns in a row.

■ We understand the style choice represented by this comment, however constrained by the word limit in the abstract, we feel that this is the clearest way to present the study. We have changed the text to landslide dam-formation susceptibility to better illustrate the compound nouns.

20: "represented as"  $\rightarrow$  "estimated from".

■ The suggested rewording is not quite correct, which speaks to the difficulty of presenting the methodology which we have expanded, see response to the general comments. We are not using the distribution to estimate a landslide volume, but rather integrating possible outcomes across the distribution, thus using it to represent landslide volumes.

20: "verify" is rather "validate"?

Yes, changed to validate.

22: "correlates" → "correlate"?

■ The high susceptibility (singular) is what correlates, so this sentence is grammatical as is.

24: How do you define "large lakes" and "low drainage areas".

■ These are defined in the appropriate sections however cannot be stated in the abstract due to the length requirement.

25: "widespread susceptibility" – Sounds a bit vague. Do you rank this susceptibility somehow by potential landslide or lake volume?

■ We have added in a quantitative assessment of the results "find that 36% of river stretches exceed a dam potential threshold..." And at this point, we feel that this level of detail and specificity is appropriate for the abstract.

26: "this hazard should be considered in the Pacific Northwest" – Reads as if this hazard is unknown there as yet. Consider rephrasing.

■ Having discussed this hazard with practitioners who are unaware of it, we feel that this phrasing is appropriate.

43: "anywhere with steep slopes" – Steepness helps, but landslides can also occur in marginally inclined terrain.

■ This is a fair point that we are unnecessarily limiting the relevance of our study. We've removed that qualifier and combined the first two sentences:" Landslides are a widespread and destructive hazard (Froude and Petley, 2018) with impacts that can cascade from slope failure to flooding when landslides intersect river valleys."

51: "are potential hazards anywhere steep slopes abut rivers" – Check grammar.

■ Altered to "Landslide dams pose hazards in locations where steep slopes abut rivers, especially in mountainous regions prone to landsliding"

58: Delete "both".

Deleted

73: "for situations where the properties of a future landslide must be inferred" – Contradicts what you state in the following "inferred from only two parameters: landslide volume and valley width" (l. 75). So is landslide volume assumed or inferred? Might pay off to define this index here.

■ We've greatly expanded this paragraph, see response to the general comment above. Earlier in the paragraph we refer to how the dam formation can be inferred from only landslide volume and valley width, while here we refer to susceptibility analysis for rivers where landslides have not happened yet, and thus properties of future landslides must be inferred. We've altered this sentence to better clarify that difference. "The MOI (Tacconi Stefanelli et al., 2016) is the simplest and the easiest to implement at landscape scales for susceptibility analysis where the properties of a future landslide must be inferred."

78: "will"  $\rightarrow$  "are likely to".

Changed. This hedging language is appropriate.

78: Delete "or dam non-formation"; this is the default situation.

■ We have deleted this extra clause.

79: Similarly, delete "formation volume and non-formation volume".

■ This sentence does refer to two separate equations so we will keep them both., Though removing the extra "volumes" to "formation and non-formation volume."

80: "satisfactorily" – Sounds a bit vague. You might want to discuss the reliability of this approach.

■ In this sentence we are referring to the study of Tacconi Stefanelli et al., (2015) where the function was hand placed to separate the datasets to the satisfaction of the authors so we feel that the vague term referring to this particular function is warranted. We agree that the reliability of such methods should be analyzed and do so for our use case in section 4.2.

90: "compound" – Is this the term you want to use? Maybe the grammar needs a check here.

Yes, we are using compound in its form as a verb which fits the usage in this case.

92: "by implementing a workflow based on" – Wordy; simply replace by "from"?

■ We think it is important to highlight that for this study the damability function is part of a larger workflow, and that removing this phrase might mislead the reader.

94: "large database of mapped landslide deposits" – From the Oregon Coast Range?

■ Yes, we've added this information in "To estimate landslide volumes we used a large database of mapped landslide deposits within the study area to fit a single empirical lognormal distribution of landslide volumes to use across the study area."

94: "define"  $\rightarrow$  "fit".

changed

95: Delete "empirical": this is what "fit" says.

■ While this word is redundant to the astute reader, we opt to keep it to increase clarity through repetition. And because we contrast this technique to deterministic techniques later in the paper.

98: "found"  $\rightarrow$  "estimated".

changed

98: "damability" – Why use quotation marks here. You had already used this term in the text.

removed

100: See previous comment.

removed

100: "are therefore useful" – Does not follow logically, and the usefulness remains to tested.

■ Fair point that testing the utility of these maps in a planning or response scenario is beyond the scope of this study and yet to be proven. So we have hedged the language a bit to reflect this: "Landslide dam susceptibility maps may be useful to planners"

104-112: This part reads like a mini-abstract. I think you can safely delete this.

■ We stand by this style choice to include a brief summary of the work including the findings at the end of the introduction.

109: "dangerously large lakes" – The literature on landslide dams indicates that not all large lakes are dangerous.

■ Fair point that using the word dangerously implies risk, and some currently existing large landslide dam lakes are not upstream population centers and appear stable. We have altered the sentence to read "large hazardous lakes".

110: Delete "investigate and".

We've deleted the redundant phrase.

120: Which part of the landslide dam do the yellow dots refer to? One yellow data sits some 70 m beside the channel in Fig. 1c.

■ The dots are placed one per dam, and not specified what part of the dam, or landslide. We did not generate this dataset and to maintain consistency with the published dataset, we have opted to not alter to published coordinates.

128: Delete either "annual" or "per year", and specify whether this an average rainfall. I think you can also safely delete "(65-200 in.)".

- We deleted 'per year'
- We will include the imperial units as those are the units that the data are reported in, and we wish to keep the paper accessible to readers local to the study area.

141-154: While I think this navigation aid has some merit, it also takes up some space. You explain all of these steps in detail below anyway. Consider trimming.

■ We added in this navigation aid to increase clarity of the methods in response to a separate reviewer from the internal USGS review and thus plan to keep it in the paper, even though is does add length.

149: "damability function (DamabilityOCR)" – Please consider a shorter abbreviation. Perhaps consider including what the function depends on.

■ We find that further abbreviations act to decrease clarity, and opt to keep the less wieldy value name.

150: "the function (DamabilityOCR-v)" – How does this differ from the function in I. 149?

■ This function is the earlier function but at each valley width the damability values are integrated across the landslide volume pdf. We have significantly edited the discussion of these equations to improve clarity.

155: What is a "non-dam inventory"?

■ We've added a brief explanation to the sentence: "(an inventory of landslides that intersect river valleys that states whether or not the landslide formed a dam),"

157: "SLIDO landslide inventory records mapped landslide deposit polygons" – How many landslides are in this inventory?

■ This is answered in the next sentence "(n>19,000)".

160: "characterize"  $\rightarrow$  "estimate".

changed

170: Please explain the two types of nodes connecting the solid lines.

We plan to edit the figure to help improve clarity and description of the differentiation of the nodes, including adding in which equation each node is representing.

175: What do you mean by "flow path uncertainties of river valleys"?

■ Where within a valley a river flows change in time, and thus are not always predictable, we have altered the sentence to read "channel size or location" instead of flow path.

178: "annual constriction ratio', 'Dimensionless Morpho-Invasion Index', and 'Dimensionless Constriction Index'" – Please provide the appropriate references directly, so that readers can chase these up. A brief explanation of these indices can help a lot here.

■ We've added more text discussing previously used methods to help place this study in context." The review of Fan et al., (2020) nicely summarizes and tests several metrics used to predict landslide dam formation. These metrics include: the 'annual constriction ratio'(ACR) which concerns the ratio between landslide velocity and valley width (Swanson et al., 1986), the 'Dimensionless Morpho-Invasion Index' (DMI) which is a more complex formulation of the ACR incorporating more physical parameters of the landslide like velocity, density, volume, and hydraulic level (Dal Sasso et al., 2014), the 'Dimensionless Constriction Index' (DCI) can also be used for stability estimates and includes parameters of the landslide like geometry, velocity, and grain size, as well as the width of the valley(Ermini and Casagli, 2003), and the 'morphological obstruction index' which concerns the ratio between log valley width and landslide volume (Tacconi Stefanelli et al., 2016). The ACR, DMI, and DCI require estimates of landslide velocity, which is difficult to measure for a known landslide, and even more difficult to estimate for an ancient or future landslides across a landscape. The MOI (Tacconi Stefanelli et al., 2016) is the simplest and the easiest to implement at landscape scales for susceptibility analysis where the properties of a future landslide must be inferred."

182: "global scale landslide dam formation susceptibility evaluation" is a wieldy term. Again, please briefly describe this method. This is really something for the introductory section (see general comment).

■ Yes this is a large unwieldy compound noun for a subject, we've cut some of the terms and by adding information on this study earlier in the introduction, the current context makes the meaning clear while the readability is improved. "Conversely, the global scale methodology of Wu et al. (2024) bypasses landslide velocity or size."

184: "sufficiently small drainage areas to capture and calibrate with landslide dams in the OCR" – This is a point that you have not demonstrated yet.

■ We've added a clause at the end of this sentence adding specificity to the statement "namely those in Struble et al., (2021)." And we've added details about the drainage area positions of these landslides to the improved background section. We feel that furth work to demonstrate this point is beyond the scope of this work.

188: "large historical dataset" – Compiling cases from?

■ We've added in "from throughout Italy" to clarify the geographic extent of the dataset. The large historical dataset is gathered from two (or more) thesis written in Italian. This level of explanation is as specific as we can get.

189: "manually placed" – Without any further constraints?

Yes. This is a major motivator for our study.

194: Delete comma.

**done**

194: "can make analysis of landslide dam likelihoods difficult" – Unclear. Is the method not designed to make this analysis easier?

■ This line has been removed with our adjustments to streamline and clarify the methods section, per the general comment suggestions.

195: "non-formation volumes" – This keeps popping up and reads as if volumes are dependent on the formation of dams. Consider rephrasing (or simply omitting) throughout.

■ Unfortunately, we need to refer to the volumes of landslides that did not form dams throughout the work. And we are reusing terms used in previous work like "non-formation volume equation". However, at the line referred to in this comment we can move away from these awkward phrases for better clarity as we have done: "This approach, can make analysis of landslide dam likelihoods difficult, requiring maps based on both the minimum volume above which a dam is possible to form and the maximum volume above which a dam is unlikely to form, and does not provide any additional uncertainty values other than a domain position"

196: "does not provide any additional uncertainty values other than a domain position" – Well, it does. You can easily estimate the misclassification from Fig. 3b by counting the false positives and false negatives, etc.

We agree that the uncertainty could be estimated based on misclassified points, and this could be done through a logistic regression, which is why this point helps to motivate our study.

198: "uncertainty in domain position" or "uncertainty at the domain position"? There is a difference between the two. Ideally, you want to estimate both together.

■ We mean the uncertainty in the domain position. We have added in a reference to the position in the named plot to help make this clear. "uncertainty in domain position in the volume-valley width plot"

189: "identify the damability function position and uncertainties" – See previous comment. What do you mean by position? The output of a logistic regression is a probability of success (or class membership).

■ By position we mean the position in the volume-valley width plot, which we have now named and referred to in the text to help make this clearer (see above comment).

201: How do you define "river stretch"?

■ The referenced text has been deleted to streamline the methods section, and follow the reviewers suggestions to shorten the paper. Elsewhere we still refer to river stretches as it is a term refereeing to a continuous section of a river with similar properties. We use this vague non-technical term here intentionally as this definition could span various techniques of breaking up rivers into sections. Later we describe how we break up rivers for analysis in our workflow for this specific study

209: "at every point where a valley width measurement is made" – See previous comment. This seems to be related.

■ The referenced text has been deleted to streamline the methods section, and follow the reviewers suggestions to shorten the paper.

217: "dam-not-formed slides" – Style. Similarly, you could relabel "Dam Non-Formation Domain" to "No landslide dams formed" in Fig. 3b. Note that regression equations in Fig. 3b formally need units of the intercept; these units depend on the differing exponents.

- We have altered the awkward phrasing as follows: "...slides that didn't form dams.."
- In 3b we keep the domain names to match the original study.
- Also in 3b we present the equations as they are presented in Tacconi Stefanelli et al. (2016).

218: "defined by lack of dam-formed slides below it" – Well, several red dots lie below this line.

■ This is true, and somewhat confuses us as well. We are only presenting the methodology and dataset from another study here. This partly motivates our statistical approach to fitting the damability function. We have altered the sentence to hedge this statement, with "general lack of dam-formed..."

Fig. 3c needs more explanation in the caption. Why do you show so few landslides from the SLIDO data if they contain more than 19,000 landslides? How did you come up with the four classes A-D?

■ We have added some text to the caption to clarify that 3c shows the dam/non-dam inventory and not all the data. "The dam/non-dam inventory for the Oregon Coast Range". We agree that this information is useful for understanding the figure however that information takes too long to describe to fit in a figure caption which is why we've included a pointer to the section where the inventory is described, where the above questions are answered "(described in section 3.4.1.)"

233: "than restriction to a GIS program" – Unclear, please elaborate.

■ We altered the text for clarity: "rather than difficult to automate GIS workflows"

234: "the ease of use in comparison to Python based codes available" – Difficult to tell because you did not demonstrate this yet.

■ "Ease of use" is a subjective qualifier for computational methodologies, we feel it's necessary to point out because it is one of the primary reasons that the cited code was not used in this analysis.

241: "capturing the prominent drainage area positions of the mapped landslide dams in the study area" – See general comments. So far, you have disclosed very little about landslide dams in the study area.

 Please see our response to the general comment and our longer background section. Where we have added more information on landslide dams in our study area

244: Why choose a "threshold elevation (10 m)"?

■ This is explained later in the paragraph, we've deleted the parenthetical "(10 m)" to avoid this question coming up for readers before it's explained.

250: "river meanders don't create their own valley walls" – Needs an explanation.

■ Thank you for pointing out an unclear sentence, we've deleted this unclear clause, and found it's not essential for understanding the algorithm.

265: What is the color scale along the river in Fig. 4d.

■ It is the same color scale as c, we've added this info to the caption "and the same color scale as c"

268: "size of future landslides can be estimated numerically based on physical laws" – Which physical laws do you mean?

■ We've altered the sentence to better clarify what we meant by physical laws: "The size of future landslides can be estimated numerically based on physically derived simulations of slope failure on the terrain"

277: "log-normal functions can fit landslide size inventories" – You mean log-normal "probability densities" or "probability distributions". You can fit any model: the question is how well it fits.

■ We've altered the text according to the suggestion to: "...demonstrated that lognormal probability distributions can adequately fit landslide size inventories..."

278: "capture absolute characteristic landslide sizes, while power laws only capture relative frequencies" – I do not follow. What do you mean by "absolute ... sizes"?

■ Power laws define ratios of relative frequencies of landslide sizes, though the same power law could be used to fit a set of landslides all smaller the 10 m³ or the much larger landslides in our study area, while the log-normal distribution is fit to absolute landslides sizes, not relative size frequencies. We've added the following sentence to better explain "The same power law (excluding cutoff size) could be fit to a set of landslides all under 10 m³ in volume, as well as the 106 or 106 m³ landslides in our inventory as long as the relative frequencies matched up"

280: "advantageous for working with statistical models" – Explain why. I think in the way that you refer to log-normal "functions" here makes them statistical models already.

■ We've adjusted the phrasing as follows as an explanation "...other statistical models that might require size estimates..."

284: "defined"  $\rightarrow$  "with".

 $\blacksquare$  Changed, to "has a mean ( $\mu$ ) of 4.44"

284: "defined by a mean ( $\mu$ ) of 4.44" – What are the units of this mean?

■ We've edited the text to state log m3.

285: "standard deviation ( $\sigma$ ) of 1.25 (or ~28,000 m³ plus or minus one standard deviation)" – Note that the standard deviation cannot be symmetric in volume. You seem to refer to the log-transformed volumes here.

Yes the standard deviation is not symmetric in volume since the normal distribution is on a log-transformed dataset. This is why we do not specify the value of the standard deviation in meters cubed. The value is referring to the mean and we have rearranged the sentence to make that clear. "has a mean ( $\mu$ ) of 4.44 ( $\sim \log(28,000 \text{ m}^3)$ ) and standard deviation ( $\sigma$ ) of 1.25"

286: "by the mapper" – How is the mapper? Was that you or the SLIDO team?

■ The SLIDO inventory is a compilation of several inventories mostly mapped by DOGAMI geologists. We've added more information in the background to clarify this. "The SLIDO database is a compilation of landslides that appear on published maps. Many of these landslides were mapped on digital elevation models generated from lidar by following a protocol that produces estimates of landslide characteristics including volume (Burns and Madin, 2002)."

287: "estimates or measures the slide depth and multiplies that by the slide area" – How do you estimate individual slide depths for >19,000 landslides?

■ These estimates were made at the time of mapping by the mapper. We've added more information in the background to clarify this." The SLIDO database is a compilation of landslides that appear on published maps. Many of these landslides were mapped on digital elevation models generated from lidar by following a protocol that produces estimates of landslide characteristics including volume (Burns and Madin, 2002)

291: Where is the y-axis scale for the "log-normal fit"? Please also report the fitting method as well as the goodness of fit.

- We plan to edit figure 5 to include the y axis mentioned.
- We've added in information on the fitting method to the above paragraph. "The distribution was fit using MATLAB's normfit function" and the goodness of fit is specified in the results.

295: "Spatially variable estimation of landslide volume (not implemented in final workflow)" – Consider dropping if this section is irrelevant to your final results.

■ We acknowledge that since section 3.3.2 is not used in our final workflow that it could be cut from the manuscript. We have trimmed the results figures from the manuscript to reflect this. However, we feel that this workflow is well motivated,

and that null results are important to present for future scientists approaching similar problems. Thus we opt to keep section 3.3.2 in the manuscript.

299: "feedback into the valley widths" – Unclear.

■ We've changed feedback to "affect" which should be more clear

307: "relatively effective" – Relative to? Effective in what way?

■ We've altered the language slightly "moderately effective at predicting landslide volumes" We've changed relatively to moderately.

308: "consider if such relationships hold for the OCR" – Does SLIDO consist only of earthquake-triggered landslides? That would be a prerequisite for comparison, right?

■ SLIDO does not consist of only earthquake triggered landslides, and before this study it was unclear if that was a necessary prerequisite for comparison.

309: "The parameters we used" need some justification. Why are these most relevant for predicting landslide volumes?

■ We've added in the following justification "We chose predictors shown to work in previous studies (Lombardo et al. 2021; Moreno et al. 2022),"

313: "500 m radius moving window" – Please explain this choice.

■ We've added in an explanation "(which is large enough to span from most rivers to hilltops in the study area)"

316/17: "general"  $\rightarrow$  "generalized".

Good catch, changed.

316: "linear and quadratic n=4 functions for each predictor variable" – Needs an explanation.

■ We've added in an explanation "to provide a simple and more complex but interpretable results"

322: "bi-logarithmic slide volume/valley width parameter space" – In the context of logistic regression, you would refer to this as the predictor or input space. The parameters describe the model instead of the data.

■ We have reframed the statement, instead naming the graph earlier (The Volume-Valley Width graph) and referring to the graph by name here. Thank you, we think this change is more correct and is clearer.

327: "parameter space" – This is technically incorrect. See previous comment.

■ We've edited this sentence to correct this and increase clarity "To find these landslides"

328: "largest surface area" – Why not volume?

■ Because not all mapped landslides have volume estimates, and we are reestimating volumes to generate the landslide dam/non-dam dataset, sorting by surface area allows us to test more possible landslides. Though in practice the top 50 volumes and surface areas match.

329: "Signs of past dam formation" – Refer to some key sources in the literature to support your assessment here.

Added a corresponding citation to Struble et al., 2021

338: "converted the area to volume using the area-volume scaling relationship" – This is slightly confusing. Did you not take the volumes from SLIDO data directly (l. 285)?

■ The volumes referred to here are for the landslide dam/non dam inventory which is a separate dataset from the SLIDO dataset. These landslides that are independently mapped by us with volume estimates from a different methodology than that used for the SLIDO dataset. These volumes are estimated using the surface area and a scaling relationship for volume to surface area from the referenced paper.

351: Please specify units for landslide volumes and valley widths in Equation 1. Please also report the corresponding units and definition ranges for parameters k, X0, and a. You should also note the (somewhat awkward) constraint that aW\_v must be unequal to 1. Dropping the base 10 is probably easier to read without changing the result.

- We specified units for volume and valley width. "estimated landslide volume (V, in meters cubed) to measured valley width (WV in meters)."
- We've removed the log base 10 from the equations, but left specifiers in the sentences following "Log base 10 transformed values of valley width and landslide volume were used to better model landslides and landscapes over several orders of magnitude in scale with a consistent relationship"

■ These are dimensionless fitting coefficients that are allowed to be any value that minimizes misfit. We define these parameters in text as describing the model "curve steepness, midpoint value, and scaling factor between valley width and volume, respectively.".

353: "reflecting the likelihood that a given combination of landslide volume and valley width would form a landslide dam" – Equation 1 shows no probabilistic statement in this regard. Please elaborate.

■ This equation (now #2 in the manuscript) defines a function to solve for the likelihood (i.e. probability) of dam formation from 0 to 1, which is exactly equivalent to probabilities from 0 – 100%.

354: "damability function was fit using nonlinear least squares regression" – The sigmoidal function of a logistic regression prohibits the use of least squares. Equation 1 has results limited to the unit interval, so least squares become increasingly limited and meaningless towards the bounds of this interval.

We have adjusted this section with more equations and steps to help clarify the methods. A sigmoidal, or any, functional form does not prohibit nonlinear least squares regression to fit the coefficients defining the function. You may mean that solving logistic regressions via MLE is common, but not exclusively the only method possible.

367: "test landslides in Burns et al. (2016)" – This is unclear. Please explain.

■ We've added in language clarifying that we are referencing to a validation test performed in this study. "In a validation test with a historical landslide inventory, in Burns et al. (2016), 65% of landslides were in 'high' areas, 16% in 'moderate' and 5% in 'low'. "

368: Insert "here" after "defined".

**done**

377: "Although these relationships were developed using alpine river geometries, they are likely still a good precursory estimate" – Avoid biasing your readers with these speculations. Why not show some results first.

■ This discusses a relationship between landslide size, drainage area, and dammed lake volume developed in the Alps, which we cannot test in our study area. So we are forced to speculate on whether this relationship holds for our

study area, in order to justify using it as our methodology. We have edited the text so as to state only facts and not pass precursory judgement on its utility. "We welcome future studies extending the fit of such lake size proxy relationships across various landscape types we are limited to this relationship developed on Alpine rivers at present."

383: "18,000" - I. 160 states "19,000".

Good catch, this has been corrected.

384: "dataset is well represented by a lognormal statistical distribution" – Hard to tell, especially in the center where the data could also be bimodal. In any case, please provide a measure of fit. You can delete "statistical" here.

■ Good point, We have added a quantitative analysis of the fit here "The goodness of fit was evaluated using a Kolmogorov-Smirnov with a K-S statistic of 0.03 indicating a good approximation."

385: Delete "single lognormal".

**deleted**

387: "This statistical distribution is inserted directly into the damability function described in section 4.2". Repetition. Consider deleting.

**Deleted**

388-396: See previous comments about possibly deleting, or at least trimming, this content together with Fig. 6 and Table 1 (which is not very informative anyway).

■ We have moved these figures to the supplementary information since they are not part of the workflow and may be distracting. However, we wish to keep some of the discussion of this work in the paper as we feel that null results especially for methodology are important to present.

419: Please provide the fitting errors for all model parameters. It may pay off to explain what these parameters mean here. Does landslide volume or valley width have a greater affect on making a location prone to damming? See my general comment on reformulating Equation 1.

■ We believe that demonstrating the goodness of fit through the validation section 4.2.2 is more important than providing individual errors on the three shape

coefficients in the function, as introducing these error values would overly complicate the equation presentation. We hope the clarifications and equation additions provided in response to other comments help illustrate this point.

420: You seem to mix the terms "probability" and "likelihood" freely here, although they are not the same.

■ Thank you for this point. We have changed almost all of the uses of probability to likelihood, which is what we are finding.

423: Delete "deterministically" – Your inference is based on a statistical model.

■ Fair point, the process can't be deterministic if the underlying inference is based on a statistical model. Deleted.

423: "landslide dam formation or non-formation volume"  $\rightarrow$  "the minimum landslide volume needed to dam a given valley width"?

Yes, we have altered the text to match this suggestion.

429: Please explain how you arrived at this result. I assume you set the exponent in Equation 2 to zero. For some reason, however, I failed to achieve this if plugging Equation 3 into Equation 2. For example, if I assume a 1000-m wide valley, Equation 3 predicts that the minimum volume for damming is 0.004 \* 1000 ^ 3.861, i.e. about 1.5 billion, cubic meters, which is consistent with what Fig. 7 shows (although the exponent there is 3.859 as opposed to 3.861 in Equation 3). Yet, if I plug this into the exponent of Equation 2: 2.5937 \* (log10(0.004 \* 1000 ^ 3.861) / log10(2.338 \* 1000) - 4.0168), I obtain ca. -3.35, whereas the result should be zero for a damability of 0.5. Please also make sure to carry over the regression estimates to Equation 3.

■ We really appreciate this comment from the reviewer which show their attention to detail and respect for reproducible science. We have added in significantly more information and equations to clarify this derivation. Please see our response to the general comment where we state what content we have added to make this section clearer.

433: "we compute damability, combining uncertainty in the damability function and range of expected landslide volumes resulting in Eq. 4" – Unclear how you computed this. Please elaborate.

■ See our response to the general comment where we state what content we have added to make this section clearer.

435: Equation 4 really refers to log10 of valley width, right?

■ No, When the base-10 logarithm of a parameter was used we specify that in the appropriate equation (e.g., Eq 3).

436: "Equation 4 (DamabilityOCR-V) includes both the logistic regression fit to the local landslide dam/non-dam inventory, and the lognormal volume distribution of local landslide dams." - Why introduce the "1 minus" term in Equation 4? Aim for consistency with your probabilistic interpretation of Equation 3.

- See our response to the general comment where we state what content we have added to make this section clearer
- The 1 minus term is introduced because we are fixing a range of landslide volumes (based on local data), as valley widths increase the overall landslide dam likelihood decreases. Without a "1 minus" term, the 'consistent' formulation of these equations would wrongly increase with increasing valley width and give unusable results.

439: "Damability values computed using Eq. 4, range from zero to one, and reflect the probability a landslide forms a dam in a valley of a certain width" – See previous comment.

■ Please see our response to the general comment where we state what content we have added to make this section clearer.

441: "uncertainty in the damability function" – Where is this included? Essentially, you subsume this in the parameter estimates.

■ See our response to the general comment where we state what content we have added to make this section clearer. The uncertainty comes from integrating the values across the spread of possible volume values as defined by the lognormal distribution.

445: The "SLIDO PDF" in Fig. 7 is not a PDF, at least not judging from the x-axis units. It might be good to mention the error estimates of the damability in the discussion.

■ This figure is a probability density function (pdf) of the SLIDO dataset. The x-axis on this plot is landslide volume (it is rotated 90°) as we believe is clear from the presentation.

453: "verification"  $\rightarrow$  "validation". Please keep results and interpretation separate. Save this point for later.

■ Changed.

454: "randomly withheld 12.5% of the dam forming points" – Why 12.5% only? "points" should be "landslides".

- We withheld 1/8th of the dataset (12.5%) as an arbitrary amount to allow for stable and robust model fitting with a sufficient blind prediction set (the 12.5%) to validate the model performance. This value could have been anything. In our preliminary testing of different samples this validation exercise choosing 10 or 25% withheld data gives similar results.
- We've changed point to landslides

458: "does not deviate much from the fit" – I think you can be a bit more quantitative about this.

■ Fair point, we will add in descriptors comparing specifically the V50 lines.

464: "ROC curves compare positive results (river points with a dam) and negative results (river points without a dam)." – ROC curves relate relative fractions of positive and negative predictions to various decision boundaries.

■ Thank you for the suggestion, we have altered the text including some of the language from this comment. "ROC curves relate fractions of positive results (river points with a dam) and negative results (river points without a dam) to decision boundaries."

465: "we do not have many mapped river stretches without dams so we substituted the entire population as "negative results" – You may want to mention all this info much earlier. Make sure you make clear whether your classification or whether your validation is potentially biased by imbalanced samples.

■ We have added more information regarding the initial fit of the damability function in response to the general comment. The damability function fit is not fit with an unbalanced dataset, rather the validation is completed with one. We feel that stating this here in the validation section is appropriate.

479: "Across the study area, 51% of the calculated damability values" – It would be nice to have a figure relating the distributions of damability vs. valley width or percent of river length. Something like that could make a nice summary of your model predictions.

■ This is a good suggestion, we plan to add, (or consider adding) a subfigure to figure 8, including a histogram of valley width results, damability results, and landslide dam susceptibility results.

500: Consider more contrast for the landslide-dam symbols in Fig. 9.

■ Fair suggestion, we plan to change the color choice for the landslide dam symbols.

512: The color scale in Fig. 10 tends to smooth out the order-of-magnitude variations in lake volumes.

■ This is a fair point, we plan to update the figure with a more variable color scale.

524: "and uncertain domains" – Unsure what you mean here.

■ This refers to specific terminology used in the referenced paper. We will add a pointer to figure 3b which includes the lines and domains specified.

525: "allows for a better characterization of the uncertainty of dam formation" – In more formal terms, your model allows a probabilistic estimate of how well you can recover mapped landslide dams from a combination of predictors.

Yes, and the result is a dam formation likelihood which in itself is an expression of uncertainty.

527: "makes adding other useful metrics (i.e., landslide susceptibility, or estimated dam lake volume) more straightforward" – How do you "add" these metrics?

■ See our comments on our clarified methodology. We have added in landslide susceptibility using EQ 3. We add in landslide susceptibility by multiplying the landslide dam likelihood by a transformed representation of the landslide susceptibility from the Oregon Statewide Landslide Susceptibility map.

529: "simplifies hazard visualizations" – To be fair, Fig. 7 does not look much different from Fig. 3b in is setup.

■ It's true that figure 7 does resemble Fig. 3b, however the difference is that our analysis can be propagated to a single value and plotted in map view as one map, while results obtained through figure 3b require multiple maps to show.

530: "damability function regression methodology" – Bulky term. Why not simply use "logistic regression".

■ We've altered the sentence to avoid the mentioned bulky term. "The logistic regression methodology that defines the damability function..."

533: "we have the best fit currently available" – By design, any regression model will generate the best fit.

■ This is a fair point that this statement is redundant, we have altered the text to remove it, and still highlight that more input data can improve fits "While we have fit the currently available inventory, new input can affect the damability function slope and uncertainty"

535: "underrepresented valley widths or volumes, could alter the shape of the function" – Well, they should, though this is something I cannot see for the combination of small landslides and narrow valleys (see general comments).

■ See our response to the general comment regarding the decision envelope.

536: "form of the damability function represents how efficient a slide of specific volume is at running out" – Really? Which model parameter tells you that?

■ We have added the word 'Conceptually' to the start of the paragraph to signify that we are referring to how the model represents the physical world, and not the specific parameters of the model.

539: "Pollock (2020)" – Is this PhD thesis publicly available?

Yes, and can be found by searching for the title as cited, here: https://digital.lib.washington.edu/researchworks/items/a59661cb-0ef5-4e17-bbd3-a9835effb12d

541: "coefficient of 0.0018" - Needs units.

■ This is a dimensionless coefficient we have added that info to the text "coefficient of 0.0018 (dimensionless)".

543: "landslides of a given volume have a consistently larger runout length than the width of the valley that they can dam" – In your study area?

■ This is true in the both our study area and the Italian dataset. The only two places where this has been measured, so we do not feel that a study area specific qualifier is needed. We have "than both" to the description of the line to clarify how it compares to both existing inventories.

545: "fact"  $\rightarrow$  "observation".

**Changed**

551: "has a lower Y intercept suggesting small slides can dam larger valley widths" – There is no y-intercept in log-log space. Do you refer to unit landslide volume and valley width here?

■ This section has been rewritten to remove 'y-intercept' what was intended, as you point out, is for x = 1, smaller slides can dam larger valley widths. "The Oregon damability function has a lower V threshold at  $W_V=1$ "

554: "speculate that small slides in Oregon may be dominated by long runout debris flows" – Difficult to assess for your readers without any background on lanslides in your study area. Something for the wish list for section 2.

■ We have added a bit of text discussing different slide types in the Oregon Coast Range to section 2. "Landslides within the SLIDO database range in landslide type, from shallow soil landslides to deep seated bedrock landslides, and from debris flows to earth flows to rotational and complex landslides."

547: "data gaps and outliers in the calibration slides" – Vague. Can you give some examples?

■ We have added examples: "For example, there are a couple very small damming landslides in the Italian inventory (Fig. S7-b) and our inventory has relatively few landslides with a volumes from 105-106 m3 and valley widths from 50-120 m (Fig. 7)."

547: "Local geology, geomorphology, and climate all likely control the form of the damability function" – By design, they do not. You only consider landslide volume and valley width as possible controls. What you refer to is the misfit between model and data.

■ It is true that the damability function is only fit using the volume, valley widths, and dam history from specific landslides. This statement refers to the ability of these underlying factors to control how a slide of a specific volume at a location of a specific valley width may runout and dam or not dam the river. For example if a dry climate suppressed runout distances this would alter which landslides

dam the valley and change the form of the damability function. We have changed the sentence to "likely indirectly control".

575: "underestimating the damability for the wide valleys because they may be more likely to experience a large volume landslide" – But these large landslides may be commensurately rarer (Figs. 5, 7).

■ This is true, large landslide are much rarer especially when the entire study area is considered. However, the point we are making here concerns locations where there may be more large landslides than other regions.

590: "The SLIDO inventory includes a wide range of failure styles, includes landslides which occurred at different times by different triggers on adjacent slopes, and was mapped by several different authors." – Again, this information is likely more useful further up in the text when you describe your methods. Like before, I suggest dropping this part from the study.

- We have added in further information concerning the SLIDO inventory in the study area section.
- We have made cuts to the paper removing some of the methodology and results of our attempt to predict landslide volumes as suggested. However we feel that this topic is still warranted here in the discussion section.

596: "makes estimating the volume of a possible future landslide on any given slope difficult" – See previous comment.

Again we have cut much of this analysis from this work, but feel that it this discussion is relevant here.

598: What are "hyper local properties"?

By hyper-local we mean properties that are only measurable on a specific hillslope and not possible to characterize at regional scale. We have added in "meter scale" to our example of a hyper local property, variability in cohesion or groundwater recharge, already in the sentence.

599: Delete "In reality, ".

**done**

602: "earthquake triggered landslide sizes (only maximum area per slope unit) may be exceptional when it comes to landslide size predictions" – Why should they be

exceptional? This notion mixes issues of data quality, method of analysis, and possible physical controls. Can you separate out the latter with confidence.

■ Fair point, this comment is speculative. We can not untangle why the studies using earthquake triggered landslide inventories mostly succeeded in predicting landslide volumes while our study did not. We feel that there is space for more research on this front. We have removed the sentence.

604: "not straightforward to use hillslope properties or geometry as a proxy for landslide volume in landslide dam susceptibility analyses" – And yet you do so by using valley width as a proxy.

■ Please see our earlier comments with our expanded description of the methodology. We treat valley width and landslide volume separately and integrate the likelihoods across all landslide volumes for each measured valley width. In some ways we do use valley width as a proxy for landslide dam susceptibility, this is because this methodology has been tested and shown to fit other landslide dam inventories [Tacconi Stefanelli et al. 2020].

607: "the largest possible landslides a hillside could produce, which may be the most important factor for landslide dam analyses" – Avoid undermining your results. If landslide volume is most important, what role then plays valley width? Using the two as additive (and standardized) predictors in a logistic regression will give your their relative weights in terms of damability.

- We meant the most important factor from the landslide side controlling landslide dam formation, and have changed the language used to reflect that "It is possible and intuitive that these proxies may match the largest possible landslides a hillside could produce, which may be more important than the mean landslide size for landslide dam analyses".
- See earlier comments about using landslide volume and valley width as additive predictors.

612: "Instead of proxies, or a local regression model, we are forced to use a region wide empirical approach to landslide volume estimation based on the mapped landslides within the study area" – I am not sure that I follow. Please elaborate.

■ We edited the passage to add more specific language "instead of site-specific landslide size proxies, we are forced to use a region wide empirical approach to landslide volume estimation based on the mapped landslides within the study area.."

618: "make this less applicable" – This refers to "inventory"?

■ Thanks for pointing out a lack of clarity. We've added the following: "make this power law based method less applicable than"

621: "susceptibility estimates dominated by valley width" – Refer to the appropriate Equation here; "dominated" is probably to exclusive, because you chose this predictor yourselves.

■ We've altered the wording and added in a reference to equation 4 (Now 6)". In this study we must assume that landslide volume remains unpredictable, and proceed with susceptibility estimates controlled by valley width (Eq. 4)," (now Eq. 6)

627: Section 5.1.3 is probably better suited for the end of the discussion.

■ We understand that a 'future work' section usually falls at the end of the discussion, however we have chosen to keep section 5.1.3 where it is for two reasons. 1- We want this section focused on future uses of our methodology to be grouped with 5.1.2 and 5.1.1 which discuss the methodology in detail. 2- We want to keep these discussions of the methodology 5.1 at the start of the discussion because they inform sections 5.2 (regional variations in our results) and 5.3 (A discussion of hazards).

634: "While the methodology of the landslide mapper may vary between studies, because this method only uses the inventory to define a lognormal volume distribution it is insulated from variations in inventory quality and completeness." – Not sure what you want to say here.

■ Thank you for pointing out a part of the manuscript that is unclear. We have edited the passage for clarity ". A detailed landslide inventory is required to characterize the landslide volume distribution. Though inventory quality and completeness vary, the use of a lognormal distribution insulates the characterization from small changes in estimated landslide volumes. A landslide dam/non-dam inventory is needed to refine the local damability function."

652: "small effect on spatial trends" – Trends of what?

■ We've added in "...in dam susceptibility estimates..." to state what trends we mean.

654-665: Reads like a great motivation for valley width as a predictor in your damability model. Why not use this earlier up, either in the introduction or methods part?

■ We intended this sentence to read as an outcome of the model methods rather than a motivation for them, so we have rephrased to keep clear that the relationship between valley width and landslide damability mentioned here is

due to the methods. We added in 'estimations' to help clarify this. "In general, valley width increases with increasing drainage area. This results in lower estimations of landslide dam susceptibility values at high drainage areas and at lower elevation and coastal regions (Fig. 11a)"

666-683: Similarly, much of this paragraph contains much needed detail for the study area description. Consider shifting this up.

■ We have added additional information to the study area section, but feel that the two studies we reference here that are Oregon Coast Range specific, pertain to points directly related to this discussion section on the spatial occurrence of landslide dam susceptibility and so will keep them in this section.

Fig. 11: Why not highlight the "trends" that you show here? Panel (a) has incorrect unit for drainage area.

- Thank you for the suggestion to highlight the trends we refer to in this figure to make it easier to read. We plan to find the best fitting lines and add them to the plots.
- Thank you for catching the error in Fig. 11 panel A units.

694: "high relief areas in the south of the study area do not always correlate with a separate rock type" – Not fully clear. What you mean by "separate"? How can you correlate relief as a quantitative measure with rock type as a categorical variable?

- 1st, we added language stating what exact rock types we are referring too, more specifically, including the sentence before the mentioned sentence. This does help to clarify our points, thank you for the suggestion.
- 2nd We added the term "visually" to clarify that we are discussing visual correlations rather than statistical correlations.

"Figure 12 demonstrates a correspondence between volcanic rock and higher relief (and landslide dam susceptibility) within the northern section of the study area. This is most notably exemplified by the Tillamook Volcanics around the Wilson River basin when compared to neighbouring landscapes in marine sedimentary rocks. The high relief areas in the south of the study area do not always visually correlate with a distinct rock type, since both high and low relief regions lie in marine sedimentary rocks, though inter-rock type variations in grain size and possibly erodibility may control landsliding and relief (LaHusen and Grant, 2024)."

700: "In the Himalaya" – Feels a bit out of place given that you discuss the OCR in the preceding and subsequent sentences.

■ We've added information to the start of the sentence to increase the flow from old information to new information presented related to the Himalayan dataset. "In other regions rock type has not been identified as a primary control on valley width, for example, in"

707: "Volcanic rocks may host narrower rivers" – The discussion about lithological controls (starting I. 690) is difficult to appreciate with the sparse information you provided in the study area. Again, I recommend delegating some material from here to that section.

■ We, have added some additional information on geology and geomorphology to the expanded study are section quoted below: "Landslide patterns within the study area are known to be controlled by sub geologic unit lithologic properties such as the dip or bedding thicknesses within the Tyee Formation (Roering et al., 2005; LaHusen and Grant, 2024)."

"Lithologic variance has been shown to impact geomorphology, with notable units such as the Tillamook Volcanics (North-central study area, Wilson river catchment Fig. 1) exhibiting sharper hilltop curvature than neighboring sedimentary units (Struble et al., 2024)."

However, we choose to repeat these relevant descriptions of the study area here so that it is close to the point that it is supporting for clarity for the reader.

714: "study area 4.6 to 4.4 respectively" – Check grammar. What are the units? Consider adding a formal test for difference in these means.

• We've slightly altered the phrasing to improve clarity Adding in the units. Our new sentence reads: In fact, the lognormal mean volume for volcanic rocks ( $\mu$ =4.6 in log m3) is slightly higher than for the marine sedimentary rocks ( $\mu$ =4.4 in log m3) which make up most of the study area.

719: "do not show a decrease in landslide susceptibility in regions of volcanic rocks" – Assuming these models are accurate, of course.

■ The purpose of the paragraph is to show that at a regional scale lithology doesn't play a primary role in controlling landslide size, though that might be intuitive. We are sure to mention the limitations of this assumption, and support it with our own data. This statement referring to these two other studies of landslides in the region assumes that the methodology used in the studies are accurate. Both studies also use methods to validate their models based on the landslide data available. We feel that we do not need to spend more time explaining the methods and validation

used in these studies since our results do not rest on them, rather their findings match ours.

761: "lake volumes of 1 million cubic meters are the minimum recorded lake volumes for catastrophic outburst floods" – How reliable do you think this estimate is?

■ This is a good point, because the impounded lake volume estimates from Costa and Schuster 1991 are likely not super reliable. However, these lake size estimates are the best we have, and we feel the uncertainty in the estimates and the uncertainty of our volume estimations are both likely nearly accurate to the order of magnitude. Which is why we've only used order of magnitude level analysis. We've added some hedging language to the referenced line help show this. "...lake volumes of around 1 million cubic..."

766: "Landslide dam disasters are usually triggered by exceptionally large landslides" – Avoid the pitfall of relying too much on reviews that are over three decades old.

■ This is an interesting point. And certainly something to consider. Here, we have added another citation from a more recent review paper, which also lists the larges dam break disasters and their large landslide triggers.

Fig. 13: Please show entire distributions and avoid truncation of the 95th percentile landslides. You show five colors but only two y-axes. Try to make it clearer that points refer to counts. The curves cannot all be PDFs, by the way. If they were scaled properly, you could make statements about the different likelihoods for each of the scenarios you discuss here.

■ We chose to cut off the high peak in the 95th percentile landslide frequency distribution, in order to increase the clarity of the figure, which we value more than presenting the height of the peak of that distribution.

802: "Future work" - You already discussed this earlier. Consider shortening.

■ This future work is specific to work that connects this methodology to societal risk, which we feel belongs at the end of the 'Hazard' discussion text. While earlier future work discussion considered ways to improve the methodology.

813: "not always predictable based on local geomorphic and geologic factors" – Not sure if this statement is fully supported by what you show: here you refer to the data, but not the choice or suitability of the models you chose.

We have rephrased the sentence to "We show that landslide volume is not always predictable using data driven models including local geomorphic and geologic factors" This reframes the statement to focus on the methodology used rather than the total predictability, since as mentioned we did not attempt all types of predictive methodology.

815: "can be successfully used to assess landslide dam susceptibility" – Summarize how successfully.

■ We added the following text after the above sentence to quantify the success: "Our methods separate sites with known landslide dams from all river sites with an ROC area under the curve of 0.834."

818: "visualize landslide dam formation likelihoods" – Refer to figure(s) showing this likelihood.

■ Added a reference to figures 9 and 10.

---

## Author Comment (AC2)

**Reply to reviewer #2**

Format of this document:

Comment text

Response text

"modified text snippet"

**Reviewer 2:**

This study presents a regional-scale workflow for mapping landslide dam formation susceptibility by integrating river valley width and landslide volume—an interesting and critical contribution to the field. The proposed damability function offers new and transferable knowledge of value to both scientific and engineering communities. The manuscript is well-structured and clearly written. I recommend minor revisions before publication in NHESS.

■ Thank you for this complimentary assessment of our manuscript!

**Specific Comments:**

- 1. The authors suggest the workflow is broadly applicable to other regions. It would be valuable to discuss its potential application in highly dynamic settings such as the Tibetan Mountains, where mega-scale landslide dams and outburst floods occur frequently (e.g., Zhang et al., 2024, Nature Communications, 15: 2878). In such contexts, factors like significant erosion may influence dam stability. Could the method be extended to incorporate these processes?
  - We agree that the applicability of this method across a diversity of landscapes remains to be shown, and we encourage such tests. We've added in a sentence at the end of 5.1.1 discussing this "Landslide dams occur in drier or wetter regions, areas with glacial or non-glacial geomorphic history and in areas with relatively minimal relief to the regions with extreme relief." And in 5.2 "It's unclear what the variability of results may look like if this methodology were applied to a region nearly entirely composed of high relief slopes such as the Himalaya."
- 2. Machine learning is increasingly used in large-scale hazard assessments. Please briefly discuss whether AI could be integrated into this workflow in the future, including potential benefits and limitations.

■ It's true that this is a type of problem to be approached by machine learning. In fact the Generalized Additive Model we use to attempt to predict landslide volumes is a class of machine learning models. Future studies may find success in these component parts of the damability analysis workflow. We've added in another sentence to section 5.1.3 stating this: "The damability approach presented here can be modified to use different methods for measuring valley width and estimating landslide volume, and it can be recalibrated with new data from other regions. Future studies may successfully employ machine learning based methodologies for any of these parts of the workflow."